# Near-Optimal Correlation Clustering with Privacy

**Vincent Cohen-Addad**
Google Research
cohenaddad@google.com

**Chenglin Fan**
Pennsylvania State University
fanchenglin@gmail.com

**Silvio Lattanzi**
Google Research
silviol@google.com

**Slobodan Mitrović**
UC Davis
smitrovic@ucdavis.edu

**Ashkan Norouzi-Fard**
Google Research
ashkannorouzi@google.com

**Nikos Parotsidis**
Google Research
nikosp@google.com

**Jakub Tarnawski**
Microsoft Research
jakub.tarnawski@microsoft.com

## Abstract

Correlation clustering is a central problem in unsupervised learning, with applications spanning community detection, duplicate detection, automated labeling and many more. In the correlation clustering problem one receives as input a set of nodes and for each node a list of co-clustering preferences, and the goal is to output a clustering that minimizes the disagreement with the specified nodes' preferences. In this paper, we introduce a simple and computationally efficient algorithm for the correlation clustering problem with provable privacy guarantees. Our additive error is stronger than those obtained in prior work and is optimal up to polylogarithmic factors for fixed privacy parameters.

## 1 Introduction

Clustering is a central problem in unsupervised machine learning. The goal of clustering is to partition a set of input objects so that similar objects are assigned to the same part while dissimilar objects are assigned to different parts of the partition. Clustering has been extensively studied throughout the years and many different formulations of the problem are known. In this paper we study the classic correlation clustering problem in the context of differential privacy.

In the correlation clustering problem [BBC04] one gets a graph whose vertices are the objects to be clustered and whose edges represent clustering preferences between the objects. More specifically, the input of the problem is a graph with positive and negative labels (and possibly with non-negative weights) on the edges, where positive edges represent similarities between vertices and negative edges represent dissimilarities. Then the correlation clustering objective asks to minimize the sum of (the weights of) positive edges across clusters plus the sum of (the weights) of negative edges within clusters. Thanks to its simple and elegant formulation, the problem has received much attention and it is used to model several practical applications including finding clustering ensembles [BGU13], duplicate detection [ARS09], community mining [CSX12], disambiguation tasks [KCMNT08], automated labelling [AHK$^+$09, CKP08] and many more. In this paper we focus on the most studied version of the problem where all edges have unit weight. In this case the best known algorithm [CALN22] has an approximation guarantee of $1.994$, which improves over a $2.06$-approximation due to [CMSY15] and a $2.5$-approximation due to [ACN08]. Furthermore, when the number of clusters is upper-bounded by $k$, a polynomial-time approximation scheme is known [GG05]. In the weighted case a $O(\log n)$-approximation is known [DEFI06], and improving

36th Conference on Neural Information Processing Systems (NeurIPS 2022).

upon this would lead to a better approximation algorithm for the notoriously difficult multicut problem. For the maximization version of the problem, where the goal is to maximize the sum of (the weights of) the positive edges within clusters plus the sum of (the weights of) the negative edges across clusters, [CGW05, Swa04] gave a 0.7666-approximation algorithm for the weighted case and a PTAS is known for the unweighted case [BBC04].

*Differential Privacy (DP)* is the *de facto* standard for user privacy [DMNS06], and it is of fundamental importance to design solutions for classic unsupervised problems in this setting. In differential privacy, the focus is on datasets, $G$ and $G'$, that differ on a single entry. An algorithm $\mathcal{A}$ is $(\epsilon, \delta)$-*differentially private* if the probabilities of observing any set of possible outputs $S$ of $\mathcal{A}$ when run on two "almost identical" inputs $G$ and $G'$ are similar: $\Pr[\mathcal{A}(G) \in S] \leq e^{\epsilon} \cdot \Pr[\mathcal{A}(G') \in S] + \delta$. Over the last decade, there have been many works considering problems related to private graphs, e.g., [HLMJ09, RHMS09, GLM+10, GRU12, BBDS13, KRSY11, KNRS13, BNSV15, AU19, US19, BCSZ18, EKKL20, BEK21, NV21, FHS21]. We briefly review two possible definitions of privacy in graphs.

**Edge Privacy.** In the edge privacy setting, two datasets are considered to be close if they differ on a single edge. [HLMJ09] introduced a differentially edge-private algorithm for releasing the degree distribution of a graph. They also proposed the notion of differential node privacy and highlighted some of the difficulties in achieving it. [GRU12] showed how to answer cut queries in a private edge model. [BBDS12] improved the error for small cuts. [GLM+10] showed how to privately release a cut close to the optimal error size. [AU19] studied the private sparsification of graphs, which was exemplified by a proposed graph meta-algorithm for privately answering cut-queries with improved accuracy. [EKKL20] studied the problem of private synthetic graph release while preserving all cuts. Recently, [NV21, FHS21] proposed frameworks for the private densest subgraph problem.

**Node Privacy.** Node differential privacy requires the algorithm to hide the presence or absence of a single node and the (arbitrary) set of edges incident to that node. However, node-DP is often difficult to achieve without compromising accuracy, because even very simple graph statistics can be highly sensitive to adding or removing a single node [KNRS13, BBDS13, BCSZ18].

Differentially private clustering has been extensively studied [BDL+17, CNX20, GKM20, CGKM21, LS20]. Nevertheless, until very recently no theoretical results were known for differentially private correlation clustering. In this context, the privacy is on the edges of the graph: namely, two graphs on the same set of vertices are *adjacent* if they differ by exactly one edge.

In a recent work, [BEK21] obtained the first differentially private correlation clustering algorithm with approximation guarantees using differentially private synthetic graph release [GRU12, EKKL20]. The framework proposed by [BEK21] is very elegant and general as it allows one to run any non-private correlation clustering approximation algorithm on a modified version of the input graph that ensures privacy. Namely, any $\alpha$-approximation algorithm to correlation clustering leads to a differentially private approximation algorithm with multiplicative approximation $\alpha$ and additive approximation $O(n^{1.75}/\epsilon)$. This applies to the more general weighted version of the problem. In the same paper, the authors also obtain an $\Omega(n/\epsilon)$ lower bound on the additive approximation of differentially private algorithms, even on unweighted graphs that consist of a single path. However, the framework from [BEK21] is rather impractical, and the additive error is far from matching the lower bound on the additive approximation. These results prompt the natural question of determining the best approximation guarantees (multiplicative and additive) that are possible under differential privacy.

As observed by [BEK21], instantiating the exponential mechanism [MT07] over the space of all clusterings yields an algorithm with additive error $O(n \log n)$. However, it is not known how to efficiently sample from the exponential mechanism for correlation clustering. [BEK21] state finding a polynomial-time algorithm that matches the additive error of the exponential mechanism as an "exciting open problem given the prominent position correlation clustering occupies both in theory and practice".

**Our Results and Technical Overview.** In this paper we present a new differentially private algorithm for the unweighted setting which achieves a constant multiplicative approximation and a nearly optimal $O(n \log^2 n)$ additive approximation (for fixed privacy parameters). More precisely, we show:

**Theorem 1.1.** *For any $\epsilon$ and $\delta$ there is an algorithm for min-disagree correlation clustering on unweighted complete graphs that is $(\epsilon, \delta)$-differentially private and returns a solution of cost $O(1) \cdot$* OPT $+ O\left(\frac{n \log^2 n \log(1/\delta)}{\epsilon^2}\right)$.

Our algorithm is given in Section 3 as Algorithm 1. Its privacy is proved in Section 4 (Theorem 4.5), and Section 5 is devoted to the approximation guarantees (Theorem 5.5).

The lower bound on the additive approximation in [BEK21] does not preclude an $(\epsilon, \delta)$-DP algorithm for correlation clustering on complete graphs with sublinear error. More precisely, [BEK21] show an $\Omega(n/\epsilon)$ lower bound for weighted paths and an $\Omega(n)$ lower bound for unweighted paths – both non-complete graphs – against pure $(\epsilon, 0)$-DP algorithms. Nevertheless, we prove that linear error is necessary even for complete unweighted graphs and $(\epsilon, \delta)$-privacy, showing that our algorithm is indeed near-optimal.

**Theorem 1.2.** *Any $(\epsilon, \delta)$-differentially private algorithm for correlation clustering on unweighted complete graphs has additive error $\Omega(n)$, assuming $\epsilon \leq 1$ and $\delta \leq 0.1$.*

The proof of Theorem 1.2 is given in Appendix C.

Our approach follows the recent result of [CALM$^+$21] for obtaining the first $O(1)$-rounds, $O(1)$-approximation algorithm for correlation clustering in the parallel setting. However, to obtain our bounds in the context of differential privacy we introduce several critical new ideas.

At a high level, the algorithm of [CALM$^+$21] trims the input graph in two steps. First, it only keeps the positive edges between those vertices that have very similar neighborhoods of positive edges (such pairs of vertices are said to be *in agreement*). More precisely, for two nodes to be in agreement the size of the intersection of the positive neighborhoods should be larger than some threshold $\beta$ times the positive degrees of each endpoint. Second, it removes the positive edges whose endpoints have lost a significant fraction of its positive neighbors during the first step (such a vertex is called a *light* vertex). Finally, the resulting clusters are given by the connected components induced by the remaining positive edges.

Our approach consists in making each of the above steps (agreement computation, light vertex computation, connected components) differentially private.

A natural way to make the agreement computation differentially private is to add Laplace noise to the size of the intersection of the neighborhoods for each pair of vertices $u, v$ and to decide that $u, v$ are in agreement if the noisy intersection size is larger than $\beta$ times the positive degrees of $u$ and $v$. One of the crucial challenges here is to make sure that the total amount of noise needed to make the entire procedure differentially private is bounded, so that we can still obtain strong approximation guarantees.

The second step, the computation of light vertices, can be made differentially private in a very natural way: simply add Laplace noise to the degree of each vertex after the removal of the edges whose endpoints are not in agreement and decide whether a vertex is light based on the noisy degree.

The third step, the connected components computation, is the most challenging. Here we need to argue that computing connected components of the graph induced by the positive edges is differentially private. In other words, we have to show that the graph induced by these edges has no "bridge"; that is, there is not a single positive edge whose removal would increase the number of connected components. This is done by establishing new properties of the algorithm and showing that if all the previous steps have succeeded, the presence of a bridge is in fact a very unlikely event. Moreover, to guarantee privacy, we must carefully modify the way we treat light vertices, as well as those of low degree.

**Discussion of Recent Work.** We note that the work of [CALM$^+$21] has been followed by the work of [AW22], who improved it in the context of streaming algorithms; however, it is not clear that there would be a benefit in using the framework of [AW22], whether in terms of running time, multiplicative, or additive approximation.

In concurrent and independent work, Liu [Liu22] proposed an $(\epsilon, \delta)$-DP algorithm that achieves a multiplicative approximation of $O(\log n)$ and an additive error of $\tilde{O}(\frac{n^{1.5} \log^2(n/\delta)}{\sqrt{\epsilon}})$ for general weighted graphs (assuming constant $\epsilon$ and $\delta$), improving upon the $O(\frac{n^{1.75} \log(n/\delta)}{\epsilon})$ error of [BEK21].

This result arises via a more careful analysis of differentially private synthetic graph release in the case of correlation clustering. Furthermore, for unweighted complete graphs Liu obtained an algorithm with constant multiplicative approximation and an additive error of $O(n \log^4 n \cdot \sqrt{\Delta^* + \log n})$ for fixed privacy parameters, where $\Delta^*$ is the maximum degree of positive edges in the graph. The latter is a pivot-based algorithm augmented with Laplace noise. This results in an $O(n^{1.5} \log^4 n)$ worst-case additive error. Our algorithm yields an additive error of $O(n \log^2 n)$ (Theorem 1.1), which significantly improves upon the result of Liu.

## 2 Preliminaries

**Correlation clustering.** In the min-disagree variant of the correlation clustering problem in complete graphs one receives as input a complete signed graph $G = (V, E^+, E^-)$, where $E^+$ (resp., $E^-$) denotes the set of "+" edges (resp., "-"), and the objective is to compute a clustering $\mathcal{C} = \{C_1, \ldots, C_t\}$ of $V$ of minimum cost, which is the number of "-" edges whose endpoints are part of the same cluster plus the number of "+" edges whose endpoints belong to the distinct clusters.

In the sequel we will use $G = (V, E)$ to refer to $G = (V, E^+, E^-)$ (with $E = E^+$).

**Differential privacy.** The definition of differential privacy (DP) is due to [DMNS06]; we use the precise formulation introduced by [Dwo06].

**Definition 2.1** (Differential Privacy). *A mechanism (randomized algorithm) $M$ with domain $\mathbb{G}$ and range $\mathcal{M}$, which we will write as $M : \mathbb{G} \to \mathcal{M}$, is $(\epsilon, \delta)$-differentially private if for any two adjacent datasets $G, G' \in \mathbb{G}$ and set of outcomes $S \subseteq \mathcal{M}$ we have*

$$\Pr[M(G) \in S] \le e^\epsilon \cdot \Pr[M(G') \in S] + \delta. \tag{1}$$

Recall that in this work, two datasets (graphs) are adjacent if they have the same set of vertices and differ by one edge.

An important property of differential privacy is that we can compose multiple differentially private subroutines into a larger DP algorithm with privacy guarantees.

**Lemma 2.2** ([DR+14], Theorem B.1). *Let $M_1 : \mathbb{G} \to \mathcal{M}_1$ be a randomized algorithm that is $(\epsilon_1, \delta_1)$-DP. Further let $M_2 : \mathbb{G} \times \mathcal{M}_1 \to \mathcal{M}_2$ be a randomized algorithm such that for every fixed $m_1 \in \mathcal{M}_1$, the mechanism $\mathbb{G} \ni G \mapsto M_2(G, m_1) \in \mathcal{M}_2$ is $(\epsilon_2, \delta_2)$-DP. Then the composed mechanism $\mathbb{G} \ni G \mapsto M_2(G, M_1(G)) \in \mathcal{M}_2$ is $(\epsilon_1 + \epsilon_2, \delta_1 + \delta_2)$-DP.*

We also use the following property to analyse the privacy of our algorithm. Its proof can be found in Appendix A.1.

**Lemma 2.3.** *Let $M_1 : \mathbb{G} \to \mathcal{M}_1$ be a randomized algorithm that is $(\epsilon, \delta)$-DP. Suppose $B \subseteq \mathcal{M}_1$ is a set of "bad outcomes" with $\Pr[M_1(G) \in B] \le \delta^*$ for any $G \in \mathbb{G}$. Further let $M_2 : \mathbb{G} \times \mathcal{M}_1 \to \mathcal{M}_2$ be a deterministic algorithm such that for every fixed "non-bad" $m_1 \in \mathcal{M}_1 \setminus B$ we have $M_2(G, m_1) = M_2(G', m_1)$ for adjacent $G, G' \in \mathbb{G}$. Then the composed mechanism $\mathbb{G} \ni G \mapsto M_2(G, M_1(G)) \in \mathcal{M}_2$ is $(\epsilon, \delta + \delta^*)$-DP.*

If we have $k$ mechanisms that are $(\epsilon, \delta)$-DP, their (adaptive) composition is $(k\epsilon, k\delta)$-DP. However, it is also possible to reduce the linear dependency on $k$ in the first parameter to roughly $\sqrt{k}$ by accepting higher additive error.

**Theorem 2.4** (Advanced Composition Theorem [DRV10]). *For all $\epsilon, \delta' \ge 0$, an adaptive composition of $k$ $(\epsilon, 0)$-differentially private mechanisms satisfies $(\epsilon', \delta')$-differential privacy for*

$$\epsilon' = \sqrt{2k \ln(1/\delta')}\epsilon + k\epsilon(e^\epsilon - 1).$$

Let $\mathrm{Lap}(b)$ be the Laplace distribution with parameter $b$ and mean 0. We will use the following two properties of Laplacian noise.

**Fact 2.5.** *Let $Y \sim \mathrm{Lap}(b)$ and $z > 0$. Then*

$$\Pr[Y > z] = \frac{1}{2} \exp\left(-\frac{z}{b}\right) \qquad \text{and} \qquad \Pr[|Y| > z] = \exp\left(-\frac{z}{b}\right).$$

**Theorem 2.6** ([DR+14], Theorem 3.6). *Let $f : \mathbb{G} \to \mathbb{R}^k$ be a function. Denote by $\Delta f$ its $\ell_1$-sensitivity, which is the maximum value of $\|f(G) - f(G')\|_1$ over adjacent datasets $G, G' \in \mathbb{G}$. Then $f + (Y_1, ..., Y_k)$, where the variables $Y_i \sim \mathrm{Lap}(\Delta f/\epsilon)$ are iid, is $(\epsilon, 0)$-DP.*

**Graph notation.** Given an input graph $G$ and a vertex $v$, we denote its set of neighbors by $N(v)$ and its degree by $d(v) = |N(v)|$. As in [CALM$^+$21], we adopt the convention that $v \in N(v)$ for every $v \in V$ (one can think that we have a self-loop at every vertex that is not removed at any step of the algorithm).

## 3 Private Algorithm for Correlation Clustering

In this section we formally define our algorithm, whose pseudo code is available in Algorithm 1 together with Definition 3.1. The algorithm uses a number of constants, which we list here for easier reading and provide feasible settings for their values:

- $\epsilon > 0$ and $\delta \in (0, \frac{1}{2})$ are user-provided privacy parameters.
- $\beta$ and $\lambda$, used the same way as in [CALM$^+$21], parametrize the notions of agreement (Definition 3.1) and lightness (Line 3), respectively. For the privacy analysis, any $\beta, \lambda \leq 0.05$ would be feasible. These parameters also control the approximation ratio, which is $O(1/(\beta\lambda))$ assuming that $\beta, \lambda$ are small enough; as in [CALM$^+$21], one can set e.g. $\beta = \lambda = 0.8/36 \approx 0.02$.
- $\epsilon_{\mathrm{agr}}, \delta_{\mathrm{agr}}$ and $\gamma$ are auxiliary parameters that control the noise used in computing agreements; they are functions of $\epsilon$ and $\delta$ and are defined in Definition 3.1.
- $\beta'$ and $\lambda'$ are used in the privacy analysis; both can be set to $0.1$.
- $T_0$ is a degree threshold; we return vertices whose (noised) degree is below that threshold as singletons, which incurs an additive loss of $O(T_0 n)$ in the approximation guarantee. We set

$$T_0 = T_1 + \frac{8\log(16/\delta)}{\epsilon},$$

where for the privacy analysis we require $T_1$ to be a large enough constant; namely, one can take the maximum of the right-hand sides of (6), (7), (8), (9), (10), (11), (14), and (15); asymptotically in terms of $\epsilon$ and $\delta$, this is of the order $O\left(\ln(1/(\epsilon\delta))^2 \ln(1/\delta)\epsilon^{-2}\right)$ (assuming $\epsilon \leq O(1)$). To additionally obtain a constant-factor approximation guarantee, we further require a polylogarithmic $T_1$, namely of the order $O(\log^2 n \log(1/\delta)\epsilon^{-2})$ – see Eq. (17) in the proof of Lemma 5.3.

The following notion is central to our algorithm and is used as part of Line 2 of Algorithm 1.

**Definition 3.1** (Noised Agreement). *Let us define $\epsilon_{\mathrm{agr}} = \epsilon/5.8$, $\delta_{\mathrm{agr}} = \delta/8$, and*

$$\gamma = \frac{\sqrt{\frac{4\epsilon_{\mathrm{agr}}}{\ln(1/\delta_{\mathrm{agr}})} + 1} + 1}{\sqrt{2}}.$$

*(Note that $\gamma \geq \sqrt{2}$.) Further let $H$ be as defined in Line 1 of Algorithm 1. For each pair of vertices $u, v \in H$, let $\mathcal{E}_{u,v}$ be an independent random variable such that*

$$\mathcal{E}_{u,v} \sim \mathrm{Lap}\left(\max\left(1, \frac{\gamma\sqrt{\max(5, d(u), d(v)) \cdot \ln(1/\delta_{\mathrm{agr}})}}{\epsilon_{\mathrm{agr}}}\right)\right).$$

*We say that two vertices $u \neq v \in H$ are in $i$-noised agreement if $|N(u) \triangle N(v)| + \mathcal{E}_{u,v} < i\beta \cdot \max(d(u), d(v))$.*

*If $u$ and $v$ are in $1$-noised agreement, we also say that $u$ and $v$ are in noised agreement; otherwise they are not in noised agreement. We say that an edge $(u, v)$ is in noised agreement if its endpoints are in noised agreement.*

Note that we do not require that $(u, v) \in E$, although Algorithm 1 will only look at agreement of edges.

## 4 Analysis of Privacy

Our analysis proceeds by fixing two adjacent datasets $G, G'$ (i.e., $G$ and $G'$ are graphs on the same vertex set that differ by one edge) and analyzing the privacy loss of each step of the algorithm. We

---

**Algorithm 1:** Private correlation clustering using noised agreement

---

**Input** : $G = (V, E)$: a graph
$\epsilon, \delta$: privacy parameters

1 Let $\hat{d}(v) = d(v) + Z_v$ denote the *noised degree* of $v$, where $Z_v \sim \mathrm{Lap}(8/\epsilon)$. Let
$H = \{v \in V : \hat{d}(v) \geq T_0\}$ denote the set of high-degree vertices.

2 Discard from $G$ the edges that are not in noised agreement (see Definition 3.1). (First compute
the set of these edges. Then remove this set. Note that this includes all edges with an endpoint
not in $H$.)

3 Let $l(v)$ be the number of edges incident to $v$ discarded in the previous step, and define
$\hat{l}(v) = l(v) + Y_v$, where $Y_v \sim \mathrm{Lap}(8/\epsilon)$. Call a vertex $v$ *light* if $\hat{l}(v) > \lambda d(v)$, and otherwise
call $v$ *heavy*.

4 Discard all edges whose both endpoints are light. Call the current graph $\hat{G}$, or the *sparsified
graph*. Compute its connected components. Output the heavy vertices in each component $C$ as
a cluster. Each light vertex is output as a singleton.

---

compose steps up to Line 3 by repeatedly invoking Lemma 2.2, which allows us to assume, when
analyzing the privacy of a step, that the intermediate outputs (states / randomness of the algorithm)
up to that point are the same between the two executions. Finally, we use Lemma 2.3 to argue that if
the intermediate outputs before Line 4 are the same, then the output of Line 4 (i.e., the final output of
Algorithm 1) does not depend on whether the input was $G$ or $G'$, except on a small fragment of the
probability space that we can charge to the additive error $\delta$.

We begin the analysis by reasoning about Line 1.

**Lemma 4.1.** *Consider Line 1 as a randomized algorithm that outputs $H$. It is $(\epsilon/4, 0)$-DP.*

*Proof.* The sensitivity $\Delta d$ of the function $d$ is 2, as adding an edge changes the degree of two vertices
by 1. Therefore, by Theorem 2.6, $\hat{d}$ is $(\epsilon/4, 0)$-DP. Furthermore, $H$ is a function that only depends
on the input (set of edges) deterministically via $\hat{d}$. □

Now we fix adjacent $G, G'$; in this proof we will think that the domain of Algorithm 1 is $\mathbb{G} = \{G, G'\}$
and the notion of "$(\cdot, \cdot)$-DP" always refers to these two fixed inputs. Denote $\{(x, y)\} = E(G) \triangle E(G')$
to be the edge on which $G$ and $G'$ differ.

**Lemma 4.2.** *Under fixed $H$, consider Line 2 as a randomized algorithm that, given $G$ or $G'$, outputs
the noised-agreement status of all edges in $E(G) \cup E(G')$. It is $(2.9\epsilon_{\mathrm{agr}}, 2\delta_{\mathrm{agr}})$-DP.*

Note that Algorithm 1 only computes the noised-agreement status of edges that are present in its
input graph (which might be the smaller of the two), but without loss of generality we can think that
it computes the status of all edges in $E(G) \cup E(G')$ (and possibly does not use this information for
the extra edge).

In the proof of Lemma 4.2 we apply the Advanced Composition Theorem to the noised-agreement
status of edges incident on $x$ or $y$ to argue that the Laplacian noise $\mathcal{E}_{u,v}$ of magnitude roughly
$\sqrt{\max(d(u), d(v))}$ is sufficient.

*Proof.* We will show that the sequence

$$(|N(u) \triangle N(v)| + \mathcal{E}_{u,v} - \beta \cdot \max(d(u), d(v)) : (u, v) \in E(G) \cup E(G')) \qquad (2)$$

(which determines the noised-agreement status of these edges) has the desired privacy guarantee.

Define $E_x$ to be those edges in $E(G) \cup E(G')$ that are adjacent to $x$, but are not $(x, y)$, and $E_y$
similarly. The sequence (2) can be decomposed into **four parts** (with independent randomness): on
$(E(G) \cup E(G')) \setminus (E_x \cup E_y)$, on $E_x$, on $E_y$, and on $(x, y)$.

For the **first part**, the function

$$(|N(u) \triangle N(v)| - \beta \cdot \max(d(u), d(v)) : (u, v) \in (E(G) \cup E(G')) \setminus (E_x \cup E_y))$$

has sensitivity 0, as for these edges we have $\{u, v\} \cap \{x, y\} = \emptyset$.

For the **second part**, we will show that the sequence

$$(|N(x)\triangle N(v)| + \mathcal{E}_{x,v} - \beta \cdot \max(d(x), d(v)) : (x,v) \in E_x) \tag{3}$$

is $(1.2\epsilon_{\mathrm{agr}}, 1.2\delta_{\mathrm{agr}})$-DP. To that end, we use the Advanced Composition Theorem (Theorem 2.4). Let $k = |E_x|$; we have $k \leq d(x)$ (note that $d$ is the degree function of the input graph, which might be the smaller of $G$, $G'$, but we also have $(x,y) \notin E_x$). Thus the sequence (3) can be seen as a composition of $k$ functions (each with independent randomness), each of which is a sum of a function $|N(x)\triangle N(v)| - \beta \cdot \max(d(x), d(v))$, which has sensitivity at most $1 + \beta$, and Laplace noise $\mathcal{E}_{x,v}$, which has magnitude at least

$$\max\left(1, \frac{\gamma\sqrt{k \cdot \ln(1/\delta_{\mathrm{agr}})}}{\epsilon_{\mathrm{agr}}}\right)$$

(where we used $\max(d(x), d(v)) \geq d(x) \geq k$). Define $\epsilon_x$ to be the inverse of this value, i.e.,

$$\epsilon_x = \min\left(1, \frac{\epsilon_{\mathrm{agr}}}{\gamma\sqrt{k \cdot \ln(1/\delta_{\mathrm{agr}})}}\right).$$

Thus by Theorem 2.6 each of the $k$ functions is $((1+\beta)\epsilon_x, 0)$-DP. By Theorem 2.4, sequence (3) is $((1+\beta)\epsilon', \delta_{\mathrm{agr}})$-DP, where

$$\begin{aligned}
\epsilon' &= \sqrt{2k\ln(1/\delta_{\mathrm{agr}})}\epsilon_x + k\epsilon_x(e^{\epsilon_x(1+\beta)} - 1) \\
&\leq \frac{\sqrt{2} \cdot \epsilon_{\mathrm{agr}}}{\gamma} + 2k\epsilon_x^2 \\
&\leq \frac{\sqrt{2} \cdot \epsilon_{\mathrm{agr}}}{\gamma} + 2k\frac{\epsilon_{\mathrm{agr}}^2}{\gamma^2 k \ln(1/\delta_{\mathrm{agr}})} \\
&= \epsilon_{\mathrm{agr}},
\end{aligned}$$

where for the first inequality we used that $e^{\epsilon_x(1+\beta)} - 1 \leq 2\epsilon_x$ for $\epsilon_x \in [0,1]$ since $\beta \leq 0.05$, and the last equality follows by substituting the value of $\gamma$ (see Definition 3.1) and reducing; our setting of $\gamma$ is in fact obtained by solving the quadratic equation $\frac{\sqrt{2}\cdot\epsilon_{\mathrm{agr}}}{\gamma} + \frac{2\epsilon_{\mathrm{agr}}^2}{\gamma^2 \ln(1/\delta_{\mathrm{agr}})} = \epsilon_{\mathrm{agr}}$.

Finally, we have $1 + \beta \leq 1.2$, and thus the sequence (3) is $(1.2\epsilon_{\mathrm{agr}}, \delta_{\mathrm{agr}})$-DP.

The **third part** is analogous to the second.

For the **fourth part**, the sensitivity of the function $|N(x)\triangle N(y)| - \beta \cdot \max(d(x), d(y))$ is $2 + \beta$ (when edge $(x,y)$ is added, $x$ and $y$ disappear from $N(x)\triangle N(y)$, and $\max(d(x), d(y))$ increases by 1). The Laplace noise $\mathcal{E}_{x,y}$ yields $(\epsilon^*, 0)$-differential privacy with

$$\epsilon^* \leq \frac{(2+\beta)\epsilon_{\mathrm{agr}}}{\gamma\sqrt{\max(5, d(x), d(y)) \cdot \ln(1/\delta_{\mathrm{agr}})}} \leq \frac{2.2\epsilon_{\mathrm{agr}}}{\sqrt{2} \cdot \sqrt{5 \cdot \ln(10)}} < 0.5\epsilon_{\mathrm{agr}},$$

where we used that $\beta \leq 0.2$, $\gamma \geq \sqrt{2}$ and $\delta_{\mathrm{agr}} \leq \frac{0.5}{8} < 0.1$.

Finally, we have a composition of four mechanisms with respective guarantees $(0,0)$, $(1.2\epsilon_{\mathrm{agr}}, \delta_{\mathrm{agr}})$ (twice), and $(0.5\epsilon_{\mathrm{agr}}, 0)$, and we can conclude the proof of the lemma using Lemma 2.2. $\qquad\square$

Next we turn our attention to Line 3.

**Lemma 4.3.** *Under fixed noised-agreement status of all edges in $E(G) \cup E(G')$, consider Line 3 as a randomized algorithm that, given $G$ or $G'$, outputs the heavy/light status of all vertices. It is $(\epsilon/4, 0)$-DP.*

*Proof.* Under fixed noised-agreement status of all edges in $E(G)\cup E(G')$, the function $(l(v)-\lambda\cdot d(v) : v \in V)$ is deterministic and we can reason about its sensitivity, which is at most $2\max(\lambda, 1-\lambda) \leq 2$ (when edge $(x,y)$ is added, the degrees $d(x)$ and $d(y)$ increase by 1, and $l(x), l(y)$ possibly increase by 1 if $x$ and $y$ are not in noised agreement). Therefore the sequence $(\hat{l}(v) - \lambda \cdot d(v) : v \in V)$ is $(\epsilon/4, 0)$-DP by Theorem 2.6, and it determines the heavy/light status of all vertices. $\qquad\square$

Now we analyze the last line (Line 4).

**Theorem 4.4.** *By a* state *let us denote the noised-agreement status of all edges in $E(G) \cup E(G')$ and heavy/light status of all vertices. Under a fixed state, consider Line 4 as a* deterministic *algorithm that, given $G$ or $G'$, outputs the final clustering. Then this clustering does not depend on whether the input graph is $G$ or $G'$, except on a set of states that arises with probability at most $\frac{3}{4}\delta$ (when steps before Line 4 are executed on either of $G$ or $G'$).*

Appendix A.2 is devoted to the proof of Theorem 4.4. Here, to get a flavor of the arguments, let us showcase what happens in the case when $x$ and $y$ are both heavy. If they are not in noised agreement, we are done; otherwise, the edge $(x, y)$ impacts the final solution only if $x$ and $y$ are not otherwise connected in the sparsified graph $\tilde{G}$. However, they are in fact likely to have many common neighbors in $\tilde{G}$, as they are in noised agreement and both heavy. Indeed, if all noise were zero, this would mean that $N(x) \cap N(y)$ is a large fraction of $\max(d(x), d(y))$ and that $x$ and $y$ do not lose many neighbors in Line 2. We show that with probability $1 - O(\delta)$, all relevant noise is below a small fraction of $\max(d(x), d(y))$, and the same argument still applies. To get this, it is crucial that the agreement noise $\mathcal{E}_{u,v}$ is of magnitude only roughly $\sqrt{\max(d(u), d(v))}$ (as opposed to e.g. $\max(d(u), d(v))$).

Once we have Theorem 4.4, we can conclude:

**Theorem 4.5.** *Algorithm 1 is $(\epsilon, \delta)$-DP.*

*Proof.* We repeatedly invoke Lemma 2.2 to argue that the part of Algorithm 1 consisting of Lines 1–3 (that outputs noised-agreement and heavy/light statuses) is $(\epsilon/4, 0) + (2.9\epsilon_{\mathrm{agr}}, 2.4\delta_{\mathrm{agr}}) + (\epsilon/4, 0) = (\epsilon, \delta/4)$-DP by Lemmas 4.1 to 4.3 (recall the setting of $\epsilon_{\mathrm{agr}}$ and $\delta_{\mathrm{agr}}$ in Definition 3.1). To conclude the proof, we argue about the last step using Lemma 2.3 and Theorem 4.4, which incurs a privacy loss of $(0, \frac{3}{4}\delta)$. $\qquad \square$

## 5  Analysis of Approximation

For vectors $\overline{\beta} \in \mathbb{R}_{\geq 0}^{V \times V}$ and $\overline{\lambda} \in \mathbb{R}_{\geq 0}^{V}$, let ALG-CC$(\overline{\beta}, \overline{\lambda})$ be the algorithm from [CALM+21] that uses $\overline{\beta}_{u,v}$ to decide an agreement between $u$ and $v$ and uses $\overline{\lambda}_v$ to decide whether $v$ is light or heavy. Let ALG-CC$(\overline{\beta}, \overline{\lambda}, E_{\mathrm{rem}})$ (stated as Algorithm 2) be a variant of ALG-CC$(\overline{\beta}, \overline{\lambda})$ that at the very first step removes $E_{\mathrm{rem}}$, then executes the remaining steps, and finally (as in Algorithm 1) outputs light vertices as singleton clusters.

---

**Algorithm 2:** ALG-CC$(\overline{\beta}, \overline{\lambda}, E_{\mathrm{rem}})$, used for the approximation analysis.

**Input** : $G = (V, E)$: a graph
$\overline{\beta} \in \mathbb{R}_{\geq 0}^{V \times V}$ : agreement parameter
$\overline{\lambda} \in \mathbb{R}_{\geq 0}^{V}$ : threshold for light vertices
$E_{\mathrm{rem}}$ : a subset of edges to be removed

1 Remove the edges in $E_{\mathrm{rem}}$.
2 Discard from $G$ the edges that are not in agreement, where $u$ and $v$ are in agreement if
$|N(u) \triangle N(v)| < \overline{\beta}_{u,v} \cdot \max(d(u), d(v))$. (First compute the set of these edges. Then remove this set.)
3 Let $l(v)$ be the number of edges incident to $v$ discarded in the previous steps. Call a vertex $v$ *light*
if $l(v) > \overline{\lambda}_v d(v)$, and otherwise call $v$ *heavy*.
4 Discard all edges whose both endpoints are light. Call the current graph $\hat{G}$, or the *sparsified graph*. Compute its connected components. Output the heavy vertices in each component $C$ as a cluster. Each light vertex is output as a singleton.

---

The strategy of our proof is to map the behavior of Algorithm 1 to the that of ALG-CC$(\overline{\beta}, \overline{\lambda}, E_{\mathrm{rem}})$ for carefully chosen $\overline{\beta}, \overline{\lambda}$, and $E_{\mathrm{rem}}$. We remark that ALG-CC is **not** actually executed by our DP-approach, but it is rather a hypothetical algorithm which, when appropriately instantiated, resembles approximation guarantees of Algorithm 1. Moreover, ALG-CC has similar structure to the approach of [CALM+21], enabling us to reuse some of the results from that prior work to establish approximation

guarantees for ALG-CC. We emphasize that the approximation guarantee from prior work we build on is invoked in a black-box manner and the analysis presented in this paper is self-contained.

We begin by showing certain monotonicity properties with respect to input parameters $\overline{\beta}$ and $\overline{\lambda}$ of ALG-CC. Given two vectors $\overline{x}$ and $\overline{y}$ labeled by a set $\mathcal{S}$, we say that $\overline{x} \leq \overline{y}$ iff $\overline{x}_s \leq \overline{y}_s$ for each $s \in \mathcal{S}$.

**Lemma 5.1.** *Let $\overline{\beta^L}, \overline{\beta^U} \in \mathbb{R}_{\geq 0}^{V \times V}$ and $\overline{\lambda^L}, \overline{\lambda^U} \in \mathbb{R}_{\geq 0}^{V}$ such that $\overline{\beta^U} \geq \overline{\beta^L}$ and $\overline{\lambda^U} \geq \overline{\lambda^L}$.*

*(i) If $u$ and $v$ are in the same cluster of $\text{ALG-CC}(\overline{\beta^L}, \overline{\lambda^L}, E_{rem})$, then $u$ and $v$ are in the same cluster of $\text{ALG-CC}(\overline{\beta^U}, \overline{\lambda^U}, E_{rem})$.*

*(ii) If $u$ and $v$ are in different clusters of $\text{ALG-CC}(\overline{\beta^U}, \overline{\lambda^U}, E_{rem})$, then $u$ and $v$ are different clusters of $\text{ALG-CC}(\overline{\beta^L}, \overline{\lambda^L}, E_{rem})$.*

The proof of Lemma 5.1 is given in Appendix B.1. We now derive the following claim that enables us to sandwich $\text{cost}(\text{ALG-CC}(\overline{\beta}, \overline{\lambda}, E_{\text{rem}}))$ between two other instances of ALG-CC.

**Lemma 5.2.** *Let $\overline{\beta^L}, \overline{\beta}, \overline{\beta^U} \in \mathbb{R}_{\geq 0}^{V \times V}$ and $\overline{\lambda^L}, \overline{\lambda}, \overline{\lambda^U} \in \mathbb{R}_{\geq 0}^{V}$ such that $\overline{\beta^U} \geq \overline{\beta} \geq \overline{\beta^L}$ and $\overline{\lambda^U} \geq \overline{\lambda} \geq \overline{\lambda^L}$. Then*

$$\text{cost}(\text{ALG-CC}(\overline{\beta}, \overline{\lambda}, E_{rem})) \leq \text{cost}(\text{ALG-CC}(\overline{\beta^U}, \overline{\lambda^U}, E_{rem})) + \text{cost}(\text{ALG-CC}(\overline{\beta^L}, \overline{\lambda^L}, E_{rem})).$$

*Proof.* We first upper-bound the cost of $\text{ALG-CC}(\overline{\beta}, \overline{\lambda}, E_{\text{rem}})$ incurred by "-" edges. If a "-" edge $\{u, v\}$ adds to the cost of clustering, it is because $u$ and $v$ are in the same cluster. By Lemma 5.1 ((i)), if $u$ and $v$ are in the same cluster of $\text{ALG-CC}(\overline{\beta}, \overline{\lambda}, E_{\text{rem}})$, then they are in the same cluster of $\text{ALG-CC}(\overline{\beta^U}, \overline{\lambda^U}, E_{\text{rem}})$ as well. Hence, the cost of $\text{ALG-CC}(\overline{\beta}, \overline{\lambda}, E_{\text{rem}})$ incurred by "-" edges is upper-bounded by $\text{cost}(\text{ALG-CC}(\overline{\beta^U}, \overline{\lambda^U}, E_{\text{rem}}))$.

In a similar way, we upper-bound the cost of $\text{ALG-CC}(\overline{\beta}, \overline{\lambda}, E_{\text{rem}})$ incurred by "+" edges by $\text{cost}(\text{ALG-CC}(\overline{\beta^L}, \overline{\lambda^L}, E_{\text{rem}}))$. If a "+" edge $\{u, v\}$ adds to the cost of clustering, it is because $u$ and $v$ are in different clusters. By Lemma 5.1 ((ii)), if $u$ and $v$ are in different cluster of $\text{ALG-CC}(\overline{\beta}, \overline{\lambda}, E_{\text{rem}})$, then they are in different clusters of $\text{ALG-CC}(\overline{\beta^L}, \overline{\lambda^L}, E_{\text{rem}})$ as well. Hence, the cost of the output of $\text{ALG-CC}(\overline{\beta}, \overline{\lambda}, E_{\text{rem}})$ incurred by "+" edges is upper-bounded by $\text{cost}(\text{ALG-CC}(\overline{\beta^L}, \overline{\lambda^L}, E_{\text{rem}}))$. $\square$

We now analyze the effect of removing edges incident to vertices which are not in $H$ defined on Line 1 of Algorithm 1. To simplify the analysis, we first ignore the step that outputs light vertices as singletons (Line 4 of Algorithm 1 and Line 4 of Algorithm 2). For a threshold $T \in \mathbb{R}_{\geq 0}$, let $E_{\leq T}$ a *subset* of edges incident to vertices of degree at most $T$.

**Lemma 5.3.** *Let Algorithm 1' be a version of Algorithm 1 that does not make singletons of light vertices on Line 4. Assume that $5\beta + 2\lambda < 1/1.1$ and also assume that $\beta$ and $\lambda$ are positive constants. With probability at least $1 - n^{-2}$, Algorithm 1' provides a solution which has $O(1)$ multiplicative and $O\left(n \cdot \left(\frac{\log n}{\epsilon} + \frac{\log^2 n \cdot \log(1/\delta)}{\min(1,\epsilon^2)}\right)\right)$ additive approximation.*

The proof of Lemma 5.3 is given in Appendix B.2. Lemma 5.3 does not take into account the cost incurred by making singleton-clusters from the light vertices, as performed on Line 4 of Algorithm 1. The next claim, whose proof is deferred to Appendix B.3, upper-bounds that cost as well.

**Lemma 5.4.** *Consider all lights vertices defined in Line 4 of Algorithm 1. Assume that $5\beta + 2\lambda < 1/1.1$. Then, with probability at least $1 - n^{-2}$, making as singleton clusters any subset of those light vertices increases the cost of clustering by $O(\text{OPT}/(\beta \cdot \lambda)^2)$, where $\text{OPT}$ denotes the cost of the optimum clustering for the input graph.*

Combining Lemmas 5.3 and 5.4, we derive our final approximation guarantee.

**Theorem 5.5.** *Assume that $5\beta + 2\lambda < 1/1.1$ and also assume that $\beta$ and $\lambda$ are positive constants. Then, with probability at least $1 - n^{-2}$*

$$\text{cost}(Algorithm\ 1) \leq O(\text{OPT}) + O\left(n \cdot \left(\frac{\log n}{\epsilon} + \frac{\log^2 n \cdot \log(1/\delta)}{\min(1,\epsilon^2)}\right)\right).$$

# 6 Future Work

It is natural to ask whether our algorithm is scalable and whether it might achieve good approximation for real-world datasets. To begin with, one might worry that our theoretical upper bound on the multiplicative approximation ratio is a large constant. However, the practical approximation performance of the algorithm [CALM+21], which we build upon, has been shown to be very good (superior to pivot-based algorithms), even in regimes of parameters $\beta$ and $\lambda$ where the theoretical bounds are not guaranteed to hold. Moreover, our algorithm is efficient and easy to implement.

Unfortunately, the additive approximation term is problematic. Recall that we return as singletons all vertices whose (noised) degree is below a threshold $T_0$. To get our theoretical bounds on approximation, we require $T_0$ to be polylogarithmic in $n$. Motivated by the experiments of [CALM+21], we could forgo these bounds in lieu of hoping for good practical performance. However, we cannot proceed similarly with respect to privacy; and $T_0$ still needs to be a large constant in order to obtain our privacy guarantees (see Section 3 for more details on $T_0$). Due to this, for most settings of $\epsilon$ and $\delta$, unless the dataset is very dense, the output of our algorithm will be close to that of the trivial $(0,0)$-DP algorithm that returns only singleton clusters.

We remark that the other known works on differentially private correlation clustering also do not provide experimental evaluations, and they seem difficult to use in practice. The algorithm [BEK21] involves approximately solving an exponential-size linear program using semidefinite programming inside a separation oracle. Liu [Liu22], similarly to us, outputs as singletons all vertices with degree below a large polylogarithmic threshold. We believe that designing a practically efficient and scalable algorithm for differentially private correlation clustering is a compelling open research direction.

On the theoretical side, it would be interesting to see whether our additive approximation bound can be improved in terms of the $\log^2 n$ factor, as well as the quadratic dependence on $1/\epsilon$. Also, perhaps our lower bound (Theorem 1.2) can be strengthened, e.g. to $\Omega(n/\epsilon)$ as in [BEK21]. Furthermore, as [CALM+21] can be implemented in $O(1)$ parallel (MPC) rounds, it is natural to ask whether this approach can lead to an algorithm that is both parallel and private. Finally, we do not address the general weighted setting (nor the maximum-agreement version) of correlation clustering in this paper; in particular, finding a DP algorithm with optimal additive guarantee for weighted inputs is an interesting follow-up question.

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
