# A Missing privacy proofs

## A.1 Proof of Lemma 2.3

We restate the lemma for convenience.

**Lemma 2.3.** *Let $M_1 : \mathbb{G} \to \mathcal{M}_1$ be a randomized algorithm that is $(\epsilon, \delta)$-DP. Suppose $B \subseteq \mathcal{M}_1$ is a set of "bad outcomes" with $\Pr[M_1(G) \in B] \leq \delta^*$ for any $G \in \mathbb{G}$. Further let $M_2 : \mathbb{G} \times \mathcal{M}_1 \to \mathcal{M}_2$ be a deterministic algorithm such that for every fixed "non-bad" $m_1 \in \mathcal{M}_1 \setminus B$ we have $M_2(G, m_1) = M_2(G', m_1)$ for adjacent $G, G' \in \mathbb{G}$. Then the composed mechanism $\mathbb{G} \ni G \mapsto M_2(G, M_1(G)) \in \mathcal{M}_2$ is $(\epsilon, \delta + \delta^*)$-DP.*

The proof is routine:

*Proof.* Fix $G, G' \in \mathbb{G}$ and a set of outcomes $S_2 \subseteq \mathcal{M}_2$. Define

$$S_1^* := \{m_1 \in \mathcal{M}_1 \setminus B : M_2(G, m_1) \in S_2\} .$$

By assumption we have

$$S_1^* = \{m_1 \in \mathcal{M}_1 \setminus B : M_2(G', m_1) \in S_2\} . \tag{4}$$

Now we can write

$$
\begin{aligned}
\Pr[M_2(G, M_1(G)) \in S_2] &\leq \Pr[M_1(G) \in B] + \Pr[M_1(G) \notin B \text{ and } M_2(G, M_1(G)) \in S_2] \\
&\leq \delta^* + \Pr[M_1(G) \in S_1^*] \\
&\stackrel{\text{DP}}{\leq} \delta^* + e^\epsilon \cdot \Pr[M_1(G') \in S_1^*] + \delta \\
&\stackrel{(4)}{=} \delta^* + e^\epsilon \cdot \Pr[M_1(G') \notin B \text{ and } M_2(G', M_1(G')) \in S_2] + \delta \\
&\leq \delta^* + e^\epsilon \cdot \Pr[M_2(G', M_1(G')) \in S_2] + \delta .
\end{aligned}
$$

$\square$

## A.2 Proof of Theorem 4.4

We restate the theorem for convenience.

**Theorem 4.4.** *By a* state *let us denote the noised-agreement status of all edges in $E(G) \cup E(G')$ and heavy/light status of all vertices. Under a fixed state, consider Line 4 as a* deterministic *algorithm that, given $G$ or $G'$, outputs the final clustering. Then this clustering does not depend on whether the input graph is $G$ or $G'$, except on a set of states that arises with probability at most $\frac{3}{4}\delta$ (when steps before Line 4 are executed on either of $G$ or $G'$).*

Let us analyze how adding a single edge $(x, y)$ can influence the output of Line 4. Namely, we will show that it cannot, unless at least one of certain bad events happens. We will list a collection of these bad events, and then we will upper-bound their probability.

First, **if $x$ and $y$ are not in noised agreement**, then $(x, y)$ was removed in Line 2 and the two outputs will be the same. In the remainder we assume that $x$ and $y$ are in noised agreement. Similarly, we can assume that $x, y \in H$ (otherwise they cannot be in noised agreement).

**If $x$ and $y$ are both light**, then similarly $(x, y)$ will be removed in Line 4 and the two outputs will be the same.

**If $x$ and $y$ are both heavy**, then $(x, y)$ will survive in $\tilde{G}$. It will affect the output if and only if it connects two components that would otherwise not be connected. However, intuitively this is unlikely, because $x$ and $y$ are heavy and in noised agreement and thus they should have common neighbors in $\tilde{G}$. Below (Lemma A.3) we will show that if no bad events (also defined below) happen, then $x$ and $y$ indeed have common neighbors in $\tilde{G}$.

**If $x$ is heavy and $y$ is light**, then similarly $(x, y)$ will survive in $\tilde{G}$, and it will affect the output if and only if it connects two components that would otherwise not be connected and that each contain a heavy vertex. More concretely, we claim that if the outputs are not equal, then $y$ must have a heavy neighbor $z \neq x$ (in $\tilde{G}$) that has no common neighbors with $x$ (except possibly $y$). For otherwise:

- if $y$ has a heavy neighbor $z \neq x$ that does have a common neighbor with $x$ (that is not $y$), then $x$ and $y$ are in the same component in $\tilde{G}$ regardless of the presence of $(x, y)$,

- if $y$ has no heavy neighbor except $x$, then (as light-light edges are removed) $y$ only has at most $x$ as a neighbor and therefore $(x, y)$ does not influence the output.

Let us call such a neighbor $z$ a *bad neighbor*. Below (Lemma A.4) we will show that if no bad events (also defined below) happen, then $y$ has no bad neighbors.

Finally, **if $x$ is light and $y$ is heavy**: analogous to the previous point. We will require that $x$ have no bad neighbor, i.e., neighbor $z \neq y$ that has no common neighbors with $y$.

**Bad events.**   We start with two helpful definitions.

**Definition A.1.** *We say that a vertex $v$ is* TV-light *(Truly Very light) if $l(v) \geq (\lambda + \lambda')d(v)$, i.e., $v$ lost a $(\lambda + \lambda')$-fraction of its neighbors in Line 2.*

**Definition A.2.** *We say that two vertices $u$, $v$ TV-disagree (Truly Very disagree) if $|N(u) \triangle N(v)| \geq (\beta + \beta') \max(d(u), d(v))$.*

Recall from Section 3 that we can set $\lambda' = \beta' = 0.1$.

Our bad events are the following:

1. $x$ and $y$ TV-disagree but are in noised agreement,
2. $x$ is TV-light but is heavy,
3. the same for $y$,
4. $x \in H$ but $d(x) < T_1$,
5. the same for $y$,
6. for each $z \in N(y) \setminus \{x, y\}$:

    6a. $y$ and $z$ do not TV-disagree, and $z$ is TV-light but is heavy, (or)
    6b. $y$ and $z$ TV-disagree, but are in noised agreement.

7. similarly for each $z \in N(x) \setminus \{x, y\}$.

Recall that we can assume that $x, y \in H$, so if bad event 4 does not happen, we have

$$d(x) \geq T_1 \tag{5}$$

and similarly for $y$ and bad event 5.

**Heavy–heavy case.**   Let us denote the neighbors of a vertex $v$ in $\tilde{G}$ by $\tilde{N}(v)$; also here we adopt the convention that $v \in \tilde{N}(v)$.

**Lemma A.3.** *If $x$ and $y$ are heavy and bad events 1–5 do not happen, then $|\tilde{N}(x) \cap \tilde{N}(y)| \geq 3$, i.e., $x$ and $y$ have another common neighbor in $\tilde{G}$.*

*Proof.* Recall that we can assume that $x$ and $y$ are in noised agreement (otherwise the two outputs are equal). Since bad event 1 does not happen, $x$ and $y$ do not TV-disagree, i.e.,

$$|N(x) \triangle N(y)| < (\beta + \beta') \max(d(x), d(y)).$$

From this we get $\min(d(x), d(y)) \geq (1 - \beta - \beta') \max(d(x), d(y))$ and thus $d(x) + d(y) = \min(d(x), d(y)) + \max(d(x), d(y)) \geq (2 - \beta - \beta') \max(d(x), d(y))$ and so

$$|N(x) \triangle N(y)| < \frac{\beta + \beta'}{2 - \beta - \beta'} (d(x) + d(y)).$$

Since $x$ is heavy but bad event 2 does not happen, $x$ is not TV-light, i.e., $l(x) < (\lambda + \lambda')d(x)$. Moreover, $l(x) = |N(x) \setminus \tilde{N}(x)|$ because $x$ is heavy (so there are no light-light edges incident to it). We use bad event 3 similarly for $y$.

We will use the following property of any two sets $A$, $B$:

$$|A \cap B| = \frac{|A| + |B| - |A \triangle B|}{2} .$$

Taking these together, we have

$$
\begin{aligned}
|\tilde{N}(x) \cap \tilde{N}(y)| &\geq |N(x) \cap N(y)| - |N(x) \setminus \tilde{N}(x)| - |N(y) \setminus \tilde{N}(y)| \\
&= \frac{d(x) + d(y) - |N(x) \triangle N(y)|}{2} - l(x) - l(y) \\
&\geq \frac{1 - \beta - \beta'}{2 - \beta - \beta'}(d(x) + d(y)) - (\lambda + \lambda')(d(x) + d(y)) \\
&= \left( \frac{1 - \beta - \beta'}{2 - \beta - \beta'} - \lambda - \lambda' \right)(d(x) + d(y)) \\
&\geq 3 ,
\end{aligned}
$$

where the last inequality follows since

$$\frac{1 - \beta - \beta'}{2 - \beta - \beta'} - \lambda - \lambda' \geq \frac{1 - 0.2 - 0.1}{2} - 0.2 - 0.1 = 0.05 > 0$$

and as, by (5), we have $d(x) + d(y) \geq 2T_1$, and $T_1$ is large enough:

$$T_1 \geq \frac{1.5}{\frac{1 - \beta - \beta'}{2 - \beta - \beta'} - \lambda - \lambda'} . \tag{6}$$

$\square$

**Heavy–light case.** Without loss of generality assume that $x$ is heavy and $y$ is light. Recall that a bad neighbor of $y$ is a vertex $z \in \tilde{N}(y) \setminus \{x, y\}$ that is heavy and has no common neighbors with $x$ (except possibly $y$).

**Lemma A.4.** *If $x$ is heavy, $y$ is light, and bad events do not happen, then $y$ has no bad neighbors.*

*Proof.* Suppose that a vertex $z \in \tilde{N}(y) \setminus \{x, y\}$ is heavy; we will show that $z$ must have common neighbors with $x$.

Since $z \in \tilde{N}(y)$, we have that $y$ and $z$ must be in noised agreement (otherwise $(y, z)$ would have been removed). Since bad event 6b does not happen, $y$ and $z$ do not TV-disagree, i.e.,

$$|N(y) \triangle N(z)| < (\beta + \beta') \max(d(y), d(z))$$

which also implies that $d(z) \geq (1 - \beta - \beta')d(y)$.

Since bad event 6a does not happen, and $y$ and $z$ do not TV-disagree, and $z$ is heavy, thus $z$ is not TV-light, i.e., $l(z) < (\lambda + \lambda')d(z)$.

As in the proof of Lemma A.3, since bad events 1 and 2 do not happen, we have

$$|N(x) \triangle N(y)| < (\beta + \beta') \max(d(x), d(y)) ,$$

which also implies that $d(x) \geq (1 - \beta - \beta')d(y)$ and $l(x) < (\lambda + \lambda')d(x)$. Similarly as in that proof, we write

$$
\begin{aligned}
|\tilde{N}(x) \cap \tilde{N}(z)| &\geq |N(x) \cap N(z)| - |N(x) \setminus \tilde{N}(x)| - |N(z) \setminus \tilde{N}(z)| \\
&= \frac{d(x) + d(z) - |N(x) \triangle N(z)|}{2} - l(x) - l(z) \\
&\geq \frac{d(x) + d(z) - |N(x) \triangle N(y)| - |N(y) \triangle N(z)|}{2} - l(x) - l(z) \\
&\geq \frac{d(x) + d(z) - (\beta + \beta')(d(x) + d(z))}{2} - (\lambda + \lambda')(d(x) + d(z)) \\
&= (1 - \beta - \beta' - 2(\lambda + \lambda')) \frac{d(x) + d(z)}{2} \\
&\geq (1 - \beta - \beta' - 2(\lambda + \lambda')) \frac{d(x) + (1 - \beta - \beta')d(y)}{2} \\
&\geq (1 - \beta - \beta' - 2(\lambda + \lambda')) \frac{2 - \beta - \beta'}{2} T_1 \\
&\geq 2\,,
\end{aligned}
$$

where the second-last inequality follows as, by (5), we have $d(x), d(y) \geq T_1$, and the last inequality follows because

$$
1 - \beta - \beta' - 2(\lambda + \lambda') \geq 1 - 0.2 - 0.1 - 2 \cdot (0.2 + 0.1) \geq 0.1 > 0
$$

and $T_1$ is large enough:

$$
T_1 \geq \frac{2 \cdot 2}{(1 - \beta - \beta' - 2(\lambda + \lambda'))\,(2 - \beta - \beta')}\,. \tag{7}
$$

$\square$

**Bounding the probability of bad events.**  Roughly, our strategy is to union-bound over all the bad events.

**Fact A.5.** *Let $A, c, d \geq 0$. If $d \geq \frac{\ln\left(\frac{c/2}{\delta}\right)}{A}$, then $\frac{1}{2} \exp(-A \cdot d) \leq \frac{\delta}{c}$.*

*Proof.*  A straightforward calculation. $\square$

**Claim A.6.** *The probability of bad event 1, conditioned on bad events 4 and 5 not happening, is at most $\delta/8$.*

*Proof.*  Start by recalling that by (5), $d(x), d(y) \geq T_1$. We have that the sought probability is at most

$$
\Pr\left[\mathcal{E}_{x,y} < -\beta' \cdot \max(d(x), d(y))\right] \leq \frac{1}{2} \exp\left(-\frac{\beta' \cdot \max(d(x), d(y))}{\mathcal{E}}\right)
$$

where we use $\mathcal{E}$ to denote the magnitude of $\mathcal{E}_{x,y}$, i.e.,

$$
\mathcal{E} = \max\left(1, \frac{\gamma\sqrt{\max(d(x), d(y)) \cdot \ln(1/\delta_{\mathrm{agr}})}}{\epsilon_{\mathrm{agr}}}\right)\,.
$$

We will satisfy both

$$
\frac{1}{2} \exp\left(-\beta' \cdot \max(d(x), d(y))\right) \leq \frac{\delta}{8}
$$

and

$$
\frac{1}{2} \exp\left(-\frac{\epsilon_{\mathrm{agr}} \cdot \beta' \cdot \max(d(x), d(y))}{\gamma\sqrt{\max(d(x), d(y)) \cdot \ln(1/\delta_{\mathrm{agr}})}}\right) \leq \frac{\delta}{8}\,.
$$

For the former, by applying Fact A.5 (for $c = 8$, $A = \beta'$ and $d = \max(d(x), d(y))$) we get that it is enough to have $\max(d(x), d(y)) \geq \frac{\ln(4/\delta)}{\beta'}$, which holds when $T_1$ is large enough:

$$
T_1 \geq \frac{\ln(4/\delta)}{\beta'}\,. \tag{8}
$$

For the latter, we want to satisfy

$$\frac{1}{2} \exp\left(-\frac{\epsilon_{\text{agr}} \cdot \beta' \cdot \sqrt{\max(d(x), d(y))}}{\gamma \sqrt{\ln(1/\delta_{\text{agr}})}}\right) \leq \frac{\delta}{8}.$$

Use Fact A.5 (for $c = 8$, $A = \frac{\epsilon_{\text{agr}} \cdot \beta'}{\gamma \sqrt{\ln(1/\delta_{\text{agr}})}}$ and $d = \sqrt{\max(d(x), d(y))}$) to get that it is enough to have

$$\sqrt{\max(d(x), d(y))} \geq \frac{\ln(4/\delta) \cdot \gamma \cdot \sqrt{\ln(1/\delta_{\text{agr}})}}{\epsilon_{\text{agr}} \cdot \beta'},$$

which is true when $T_1$ is large enough:

$$T_1 \geq \left(\frac{\ln(4/\delta) \cdot \gamma}{\epsilon_{\text{agr}} \cdot \beta'}\right)^2 \cdot \ln(1/\delta_{\text{agr}}). \tag{9}$$

$\square$

**Claim A.7.** *The probability of bad event 2, conditioned on bad events 4 and 5 not happening, is at most $\delta/32$.*

*Proof.* Start by recalling that by (5), $d(x) \geq T_1$. If $x$ is TV-light but heavy, then we must have $Y_x < \lambda' \cdot d(x)$. We have that the sought probability is at most

$$\frac{1}{2} \exp\left(-\frac{\lambda' \cdot d(x) \cdot \epsilon}{8}\right)$$

and by Fact A.5 (with $c = 32$, $d = d(x)$ and $A = \frac{\lambda' \cdot \epsilon}{8}$) this is at most $\delta/32$ because $d(x) \geq T_1$ and $T_1$ is large enough:

$$T_1 \geq \frac{8 \ln(16/\delta)}{\lambda' \cdot \epsilon}. \tag{10}$$

$\square$

**Claim A.8.** *The probability of bad event 4 is at most $\delta/32$.*

*Proof.* For bad event 4 to happen, we must have $Z_x \geq T_0 - T_1 = \frac{8 \ln(16/\delta)}{\epsilon}$; as $Z_x \sim \text{Lap}(8/\epsilon)$, this happens with probability $\frac{1}{2} \exp(-\ln(16/\delta)) = \delta/32$. $\square$

The following two facts are more involved versions of Fact A.5.

**Fact A.9.** *Let $A, d \geq 0$. If $d \geq \frac{1.6 \ln\left(\frac{4}{\delta A}\right)}{A}$, then $\frac{1}{2} \exp(-A \cdot d) \leq \frac{\delta}{8d}$.*

*Proof.* We use the following analytic inequality: for $\alpha, x > 0$, if $x \geq 1.6 \ln(\alpha)$, then $x \geq \ln(\alpha x)$. We substitute $x = A \cdot d$ and $\alpha = \frac{4}{\delta A}$. Then by the analytic inequality, $A \cdot d \geq \ln\left(\frac{4d}{\delta}\right)$. Negate and then exponentiate both sides. $\square$

**Fact A.10.** *Let $A, d \geq 0$. If $\sqrt{d} \geq \frac{2.8 \cdot \left(1 + \ln\left(\frac{2}{\sqrt{\delta} A}\right)\right)}{A}$, then $\frac{1}{2} \exp(-A \cdot \sqrt{d}) \leq \frac{\delta}{8d}$.*

*Proof.* We use the following analytic inequality: for $\alpha, x > 0$, if $x \geq 2.8(\ln(\alpha) + 1)$, then $x \geq 2 \ln(\alpha x)$. We substitute $x = A\sqrt{d}$ and $\alpha = \frac{2}{\sqrt{\delta} A}$. Then by the analytic inequality, $A \cdot \sqrt{d} \geq \ln\left(\frac{4d}{\delta}\right)$. Negate and then exponentiate both sides. $\square$

**Claim A.11.** *For any $z \in N(y) \setminus \{x, y\}$, the probability of bad event 6a for $z$, conditioned on bad events 4 and 5 not happening, is at most $\frac{\delta}{8d(y)}$.*

*Proof.* The proof is similar as for Claim A.7 but somewhat more involved as $d(y)$ appears also in the probability bound.

When $z$ is TV-light but heavy, we must have $Y_z < -\lambda' \cdot d(z)$. When $y$ and $z$ do not TV-disagree, we have $d(z) \geq (1-\beta-\beta')d(y)$. Thus, if bad event 6a happens, we must have $Y_z < -\lambda' \cdot (1-\beta-\beta')d(y)$. Thus the sought probability is at most

$$\Pr\left[Y_z < -\lambda' \cdot (1 - \beta - \beta')d(y)\right] = \frac{1}{2} \exp\left(-\frac{\lambda' \cdot (1 - \beta - \beta')d(y) \cdot \epsilon}{8}\right).$$

By Fact A.9 (invoked for $d = d(y)$ and $A = \frac{\lambda' \cdot (1-\beta-\beta') \cdot \epsilon}{8}$), this is at most $\frac{\delta}{8d(y)}$ because $d(y) \geq T_1$ by (5) and $T_1$ is large enough:

$$T_1 \geq \frac{1.6 \ln\left(\frac{4 \cdot 8}{\delta \lambda' \cdot (1-\beta-\beta') \cdot \epsilon}\right) \cdot 8}{\lambda' \cdot (1 - \beta - \beta') \cdot \epsilon} \, . \tag{11}$$

$\square$

**Claim A.12.** *For any $z \in N(y) \setminus \{x, y\}$, the probability of bad event 6b for $z$, conditioned on bad events 4 and 5 not happening, is at most $\frac{\delta}{8d(y)}$.*

*Proof.* The proof is similar as for Claim A.6 but somewhat more involved as $d(y)$ appears also in the probability bound. Start by recalling that by (5), $d(y) \geq T_1$. We have that the sought probability is at most

$$\Pr\left[\mathcal{E}_{y,z} < -\beta' \cdot \max(d(y), d(z))\right] \leq \frac{1}{2} \exp\left(-\frac{\beta' \cdot \max(d(y), d(z))}{\mathcal{E}}\right)$$

where we use $\mathcal{E}$ to denote the magnitude of $\mathcal{E}_{y,z}$, i.e.,

$$\mathcal{E} = \max\left(1, \frac{\gamma\sqrt{\max(d(y), d(z)) \cdot \ln(1/\delta_{\mathrm{agr}})}}{\epsilon_{\mathrm{agr}}}\right) \, .$$

We will satisfy both

$$\frac{1}{2} \exp\left(-\beta' \cdot \max(d(y), d(z))\right) \leq \frac{1}{2} \exp\left(-\beta' \cdot d(y)\right) \leq \frac{\delta}{8d(y)} \tag{12}$$

and

$$\frac{1}{2} \exp\left(-\frac{\epsilon_{\mathrm{agr}} \cdot \beta' \cdot \max(d(y), d(z))}{\gamma\sqrt{\max(d(y), d(z)) \cdot \ln(1/\delta_{\mathrm{agr}})}}\right) \leq \frac{1}{2} \exp\left(-\frac{\epsilon_{\mathrm{agr}} \cdot \beta' \cdot \sqrt{d(y)}}{\gamma\sqrt{\ln(1/\delta_{\mathrm{agr}})}}\right) \leq \frac{\delta}{8d(y)} \, . \tag{13}$$

For the former, by applying Fact A.9 (for $A = \beta'$ and $d = d(y)$) we get that (12) holds because $d(y) \geq T_1$ and $T_1$ is large enough:

$$T_1 \geq \frac{1.6 \ln\left(\frac{4}{\delta \cdot \beta'}\right)}{\beta'} \, . \tag{14}$$

For the latter, by applying Fact A.10 (for $A = \frac{\epsilon_{\mathrm{agr}} \cdot \beta'}{\gamma\sqrt{\ln(1/\delta_{\mathrm{agr}})}}$ and $d = d(y)$) we get that (13) holds because $d(y) \geq T_1$ and $T_1$ is large enough:

$$T_1 \geq \left(\frac{2.8 \left(1 + \ln\left(\frac{2}{\sqrt{\delta}A}\right)\right)}{A}\right)^2 = \left(\frac{2.8 \left(1 + \ln\left(\frac{2\gamma\sqrt{\ln(1/\delta_{\mathrm{agr}})}}{\sqrt{\delta}\epsilon_{\mathrm{agr}} \cdot \beta'}\right)\right)\gamma\sqrt{\ln(1/\delta_{\mathrm{agr}})}}{\epsilon_{\mathrm{agr}} \cdot \beta'}\right)^2 \, . \tag{15}$$

$\square$

Now we may conclude the proof of Theorem 4.4. We use the property that if $A$, $B$ are events, then $\Pr\left[A \cup B\right] \leq \Pr\left[A\right] + \Pr\left[B \mid \text{not } A\right]$ (with $A$ being bad events 4 or 5). By Claim A.8, the probability of bad events 4 or 5 is at most $\delta/16$. Conditioned on these not happening, bad event 1 is handled by Claim A.6 and bad events 2–3 are handled by Claim A.7; these incur $\delta/8 + 2 \cdot \delta/32$, in total $\delta/4$ so far. Next, there are $d(y)$ bad events of type 6a (and the same for 6b), thus we get $2 \cdot d(y) \cdot \frac{\delta}{8d(y)} = \delta/4$ by Claims A.11 and A.12; and we get the same from bad events 7a and 7b. Summing everything up yields $\frac{3}{4}\delta$. $\blacksquare$

# B  Proofs Missing from Section 5

## B.1  Proof of Lemma 5.1

First, we prove the following claim.

**Lemma B.1.** *Let $\overline{\beta^L}, \overline{\beta^U} \in \mathbb{R}_{\geq 0}^{V \times V}$ and $\overline{\lambda^L}, \overline{\lambda^U} \in \mathbb{R}_{\geq 0}^V$ such that $\overline{\beta^U} \geq \overline{\beta^L}$ and $\overline{\lambda^U} \geq \overline{\lambda^L}$. Let $E_{rem}$ be a subset of edges. Then, the following holds:*

- *(A) If $v$ is light in $\text{ALG-CC}(\overline{\beta^U}, \overline{\lambda^U}, E_{rem})$, then $v$ is light in $\text{ALG-CC}(\overline{\beta^L}, \overline{\lambda^L}, E_{rem})$.*

- *(B) If $v$ is heavy in $\text{ALG-CC}(\overline{\beta^L}, \overline{\lambda^L}, E_{rem})$, then $v$ is heavy in $\text{ALG-CC}(\overline{\beta^U}, \overline{\lambda^U}, E_{rem})$.*

- *(C) If an edge $e$ is removed in $\text{ALG-CC}(\overline{\beta^U}, \overline{\lambda^U}, E_{rem})$, then $e$ is removed in $\text{ALG-CC}(\overline{\beta^L}, \overline{\lambda^L}, E_{rem})$ as well.*

- *(D) If an edge $e$ remains in $\text{ALG-CC}(\overline{\beta^L}, \overline{\lambda^L}, E_{rem})$, then $e$ remains in $\text{ALG-CC}(\overline{\beta^U}, \overline{\lambda^U}, E_{rem})$ as well.*

*Proof.* Observe that $|N(u) \triangle N(v)| \leq \overline{\beta^L}_{u,v} \max\{d(u), d(v)\}$ implies $|N(u) \triangle N(v)| \leq \overline{\beta^U}_{u,v} \max\{d(u), d(v)\}$ as $\overline{\beta^L}_{u,v} \leq \overline{\beta^U}_{u,v}$. Hence, if $u$ and $v$ are in agreement in $\text{ALG-CC}(\overline{\beta^L}, \overline{\lambda^L}, E_{\text{rem}})$, then $u$ and $v$ are in agreement in $\text{ALG-CC}(\overline{\beta^U}, \overline{\lambda^U}, E_{\text{rem}})$ as well. Similarly, if $u$ and $v$ are not in agreement in $\text{ALG-CC}(\overline{\beta^U}, \overline{\lambda^U}, E_{\text{rem}})$, then $u$ and $v$ are not in agreement in $\text{ALG-CC}(\overline{\beta^L}, \overline{\lambda^L}, E_{\text{rem}})$ as well. These observations immediately yield Properties (A) and (B).

To prove Properties (C) and (D), observe that an edge $e = \{u, v\}$ is removed from a graph if $u$ and $v$ are not in agreement, or if $u$ and $v$ are light, or if $e \in E_{\text{rem}}$. From our discussion above and from Property (A), if $e$ is removed from $\text{ALG-CC}(\overline{\beta^U}, \overline{\lambda^U}, E_{\text{rem}})$, then $e$ is removed from $\text{ALG-CC}(\overline{\beta^L}, \overline{\lambda^L}, E_{\text{rem}})$ as well. On the other hand, $e \notin E_{\text{rem}}$ remains in $\text{ALG-CC}(\overline{\beta^L}, \overline{\lambda^L}, E_{\text{rem}})$ if $u$ and $v$ are in agreement, and if $u$ or $v$ is heavy. Property (B) and our discussion about vertices in agreement imply Property (D).[1]  □

As a corollary, we obtain the proof of Lemma 5.1.

**Lemma 5.1.** *Let $\overline{\beta^L}, \overline{\beta^U} \in \mathbb{R}_{\geq 0}^{V \times V}$ and $\overline{\lambda^L}, \overline{\lambda^U} \in \mathbb{R}_{\geq 0}^V$ such that $\overline{\beta^U} \geq \overline{\beta^L}$ and $\overline{\lambda^U} \geq \overline{\lambda^L}$.*

- *(i) If $u$ and $v$ are in the same cluster of $\text{ALG-CC}(\overline{\beta^L}, \overline{\lambda^L}, E_{rem})$, then $u$ and $v$ are in the same cluster of $\text{ALG-CC}(\overline{\beta^U}, \overline{\lambda^U}, E_{rem})$.*

- *(ii) If $u$ and $v$ are in different clusters of $\text{ALG-CC}(\overline{\beta^U}, \overline{\lambda^U}, E_{rem})$, then $u$ and $v$ are different clusters of $\text{ALG-CC}(\overline{\beta^L}, \overline{\lambda^L}, E_{rem})$.*

*Proof.*  (i)  Consider a path $P$ between $u$ and $v$ that makes them being in the same cluster/component in $\text{ALG-CC}(\overline{\beta^L}, \overline{\lambda^L}, E_{\text{rem}})$. Then, by Lemma B.1 (D) $P$ remains in $\text{ALG-CC}(\overline{\beta^U}, \overline{\lambda^U}, E_{\text{rem}})$ as well. Hence, $u$ and $v$ are in the same cluster of $\text{ALG-CC}(\overline{\beta^U}, \overline{\lambda^U}, E_{\text{rem}})$.

(ii)  Follows from Property (i) by contraposition.

□

## B.2  Proof of Lemma 5.3

We begin by proving the following claim.

---

[1] Also, by contraposition, Property (D) follows from Property (C) and Property (B) follows from Property (A).

**Lemma B.2.** *Let* ALG-CC$'$ *be a version of* ALG-CC *that does not make singletons of light vertices on Line 4 of Algorithm 2. Let* $\overline{\beta} \in \mathbb{R}_{\geq 0}^{V \times V}$ *and* $\overline{\lambda} \in \mathbb{R}_{\geq 0}^{V}$ *be two constant vectors, i.e.,* $\overline{\beta} = \beta\overline{1}$ *and* $\overline{\lambda} = \lambda\overline{1}$. *Assume that* $5\beta + 2\lambda < 1$. *Then, it holds*

$$\mathrm{cost}(\text{ALG-CC}'(\overline{\beta}, \overline{\lambda}, E_{\leq T})) \leq O(OPT/(\beta\lambda)) + O(n \cdot T/(1 - 4\beta)^3),$$

*where OPT denotes the cost of the optimum clustering for the input graph.*

*Proof.* Consider a non-singleton cluster $C$ output by ALG-CC$'(\overline{\beta}, \overline{\lambda}, \emptyset)$. Let $u$ be a vertex in $C$. We now show that for any $v \in C$, such that $u$ or $v$ is heavy, it holds that $d(v) \geq (1 - 4\beta)d(u)$. To that end, we recall that in [CALM$^+$21] (Lemma 3.3 of the arXiv version) it was shown that

$$|N(u)\triangle N(v)| \leq 4\beta \max\{d(u), d(v)\}. \tag{16}$$

Assume that $d(u) \geq d(v)$, as otherwise $d(v) \geq (1 - 4\beta)d(u)$ holds directly. Then, from Eq. (16) we have

$$d(u) - d(v) \leq |N(u)\triangle N(v)| \leq 4\beta d(u),$$

further implying

$$d(v) \geq (1 - 4\beta)d(u).$$

Moreover, this provides a relation between $d(v)$ and $d(u)$ even if both vertices are light. To see that, fix any heavy vertex $z$ in the cluster. Any vertex $u$ has $d(u) \leq d(z)/(1 - 4\beta)$ and also $d(u) \geq (1 - 4\beta)d(z)$. This implies that if $u$ and $v$ belong to the same cluster than $d(u) \geq (1 - 4\beta)^2 d(v)$, even if both $u$ and $v$ are light.

Let $E_{\leq T}$ be a subset (any such) of edges incident to vertices with degree at most $T$. We will show that forcing ALG-CC$'$ to remove $E_{\leq T}$ does not affect how vertices of degree at least $T/(1 - 4\beta)^3$ are clustered by ALG-CC$'$. To see that, observe that a vertex $x$ having degree at most $T$ and a vertex $y$ having degree at least $T/(1 - \beta) + 1$ are not in agreement. Hence, forcing ALG-CC$'$ to remove $E_{\leq T}$ does not affect whether vertex $y$ is light or not.

However, removing $E_{\leq T}$ might affect whether a vertex $z$ with degree $T/(1 - \beta) < T/(1 - 4\beta)$ is light or not. Nevertheless, from our discussion above, a vertex $y$ with degree at least $T/(1 - 4\beta)^3$ is not clustered together with $z$ by ALG-CC$'(\beta, \lambda, \emptyset)$, regardless of whether $z$ is heavy or light.

This implies that the cost of clustering vertices of degree at least $T/(1-4\beta)^3$ by ALG-CC$'(\beta, \lambda, E_{\leq T})$ is upper-bounded by $\mathrm{cost}(\text{ALG-CC}'(\overline{\beta}, \overline{\lambda}, \emptyset)) \leq O(OPT/(\beta\lambda))$. Notice that the inequality follows since ALG-CC$'(\overline{\beta}, \overline{\lambda}, \emptyset)$ is a $O(1/(\beta\lambda))$-approximation of $OPT$ and $\beta < 0.2$.

It remains to account for the cost effect of ALG-CC$'(\overline{\beta}, \overline{\lambda}, E_{\leq T})$ on the vertices of degree less than $T/(1 - 4\beta)^3$. This part of the analysis follows from the fact that forcing ALG-CC$'$ to remove $E_{\leq T}$ only reduces connectivity compared to the output of ALG-CC$'$ without removing $E_{\leq T}$. That is, in addition to removing edges even between vertices that might be in agreement, removal of $E_{\leq T}$ increases a chance for a vertex to become light. Hence, the clusters of ALG-CC$'$ with removals of $E_{\leq T}$ are only potentially further clustered compared to the output of ALG-CC$'$ without the removal. This means that ALG-CC$'$ with the removal of $E_{\leq T}$ potentially cuts additional "+" edges, but it does not include additional "-" edges in the same cluster. Given that only vertices of degree at most $T/(1 - 4\beta)^3$ are affected, the number of additional "+" edges cut is $O(n \cdot T/(1 - 4\beta)^3)$.

This completes the analysis. $\square$

**Lemma 5.3.** *Let Algorithm 1' be a version of Algorithm 1 that does not make singletons of light vertices on Line 4. Assume that $5\beta + 2\lambda < 1/1.1$ and also assume that $\beta$ and $\lambda$ are positive constants. With probability at least $1 - n^{-2}$, Algorithm 1' provides a solution which has $O(1)$ multiplicative and $O\left(n \cdot \left(\frac{\log n}{\epsilon} + \frac{\log^2 n \cdot \log(1/\delta)}{\min(1, \epsilon^2)}\right)\right)$ additive approximation.*

*Proof.* We now analyze under which condition noised agreement and $\hat{l}(v)$ can be seen as a slight perturbation of $\beta$ and $\lambda$. That will enable us to employ Lemmas 5.2 and B.2 to conclude the proof of this theorem.

**Analyzing noised agreement.** Recall that a noised agreement (Definition 3.1) states

$$|N(u)\triangle N(v)| + \mathcal{E}_{u,v} < \beta \cdot \max(d(u), d(v)).$$

This inequality can be rewritten as

$$|N(u)\triangle N(v)| < \left(1 - \frac{\mathcal{E}_{u,v}}{\beta \cdot \max(d(u), d(v))}\right) \beta \cdot \max(d(u), d(v)).$$

As a reminder, $\mathcal{E}_{u,v}$ is drawn from $\mathrm{Lap}(C_{u,v} \cdot \sqrt{\max(d(u), d(v)) \ln(1/\delta)}/\epsilon_{\mathrm{agr}})$, where $C_{u,v}$ can be upper-bounded by $C = \sqrt{4\epsilon_{\mathrm{agr}} + 1} + 1$. Let $b = C \cdot \sqrt{\max(d(u), d(v)) \ln(1/\delta)}/\epsilon_{\mathrm{agr}}$. From Fact 2.5 we have that

$$\Pr\left[|\mathcal{E}_{u,v}| > 5 \cdot b \cdot \log n\right] \leq n^{-5}.$$

Therefore, with probability at least $1 - n^{-5}$ we have that

$$\left|\frac{\mathcal{E}_{u,v}}{\beta \cdot \max(d(u), d(v))}\right| \leq \frac{5 \cdot \log n \cdot C \cdot \sqrt{\max(d(u), d(v)) \ln(1/\delta)}}{\epsilon_{\mathrm{agr}} \cdot \beta \cdot \max(d(u), d(v))} = \frac{5 \cdot \log n \cdot C \cdot \sqrt{\ln(1/\delta)}}{\epsilon_{\mathrm{agr}} \cdot \beta \cdot \sqrt{\max(d(u), d(v))}}$$

Therefore, for $\max(d(u), d(v)) \geq \frac{2500 \cdot C^2 \cdot \log^2 n \cdot \log(1/\delta)}{\beta^2 \cdot \epsilon_{\mathrm{agr}}^2}$ we have that with probability at least $1 - n^{-5}$ it holds

$$1 - \frac{\mathcal{E}_{u,v}}{\beta \cdot \max(d(u), d(v))} \in [9/10, 11/10].$$

**Analyzing noised $l(v)$.** As a reminder, $\hat{l}(v) = l(v) + Y_v$, where $Y_v$ is drawn from $\mathrm{Lap}(8/\epsilon)$. The condition $\hat{l}(v) > \lambda d(v)$ can be rewritten as

$$l(v) > \left(1 - \frac{Y_v}{\lambda d(v)}\right) \lambda d(v).$$

Also, we have

$$\Pr\left[|Y_v| > \frac{40 \log n}{\epsilon}\right] < n^{-5}.$$

Hence, if $d(v) \geq \frac{400 \log n}{\lambda \epsilon}$ then with probability at least $1 - n^{-5}$ we have that

$$1 - \frac{Y_v}{\lambda d(v)} \in [9/10, 11/10].$$

**Analyzing noised degrees.** Recall that noised degree $\hat{d}(v)$ is defined as $\hat{d}(v) = d(v) + Z_v$, where $Z_v$ is drawn from $\mathrm{Lap}(8/\epsilon)$. From Fact 2.5 we have

$$\Pr\left[|Z_v| > \frac{40 \log n}{\epsilon}\right] < n^{-5}.$$

Hence, with probability at least $1 - n^{-5}$, a vertex of degree at least $T_0 + 40 \log n/\epsilon$ is in $H$ defined on Line 1 of Algorithm 1. Also, with probability at least $1 - n^{-5}$ a vertex with degree less than $T_0 - 40 \log n/\epsilon$ is not in $H$.

**Combining the ingredients.** Define

$$T' = \max\left(\frac{400 \log n}{\lambda \epsilon}, \frac{2500 \cdot C^2 \cdot \log^2 n \cdot \log(1/\delta)}{\beta^2 \cdot \epsilon_{\mathrm{agr}}^2}\right)$$

Our analysis shows that for a vertex $v$ such that $d(v) \geq T'$ the following holds with probability at least $1 - 2n^{-5}$:

(i) The perturbation by $\mathcal{E}_{u,v}$ in Definition 3.1 can be seen as multiplicatively perturbing $\overline{\beta}_{u,v}$ by a number from the interval $[-1/10, 1/10]$.

(ii) The perturbation of $l(v)$ by $Y_v$ can be seen as multiplicatively perturbing $\overline{\lambda}_v$ by a number from the interval $[-1/10, 1/10]$.

Let $T = T_0 + \frac{40 \log n}{\epsilon}$. Let $T_0 \geq T' + \frac{40 \log n}{\epsilon}$. Note that this imposes a constraint on $T_1$, which is

$$T_1 \geq T' + \frac{40 \log n}{\epsilon} - \frac{8 \log(16/\delta)}{\epsilon}. \tag{17}$$

Then, following our analysis above, each vertex in $H$ has degree at least $T'$, and each vertex of degree at least $T$ is in $H$. Let $E_{\leq T}$ be the set of edges incident to vertices which are not in $H$; these edges are effectively removed from the graph. Observe that for a vertex $u$ which do not belong to $H$ it is irrelevant what $\overline{\beta}_{u,\cdot}$ values are or what $\overline{\lambda}_u$ is, as all its incident edges are removed. To conclude the proof, define $\overline{\beta^L} = 0.9 \cdot \beta \cdot \overline{1}$, $\overline{\beta^U} = 1.1 \cdot \beta \cdot \overline{1}$, $\overline{\lambda^L} = 0.9 \cdot \lambda \cdot \overline{1}$, and $\overline{\lambda^U} = 1.1 \cdot \lambda \cdot \overline{1}$. By Lemma 5.2 and Properties (i) and (ii) we have that

$$\text{cost}(Algorithm\ 1') \leq \text{cost}(\text{ALG-CC}(\overline{\beta^L}, \overline{\lambda^L}, E_{\leq T})) + \text{cost}(\text{ALG-CC}(\overline{\beta^U}, \overline{\lambda^U}, E_{\leq T})).$$

By Lemma B.2 the latter sum is upper-bounded by $O(OPT/(\beta\lambda)) + O(n \cdot T/(1 - 4\beta)^3)$. Note that we replace the condition $5\beta + 2\lambda$ in the statement of Lemma B.2 by $5\beta + 2\lambda < 1/1.1$ in this lemma so to account for the perturbations. Moreover, we can upper-bound $T$ by

$$T \leq O\left( \frac{\log n}{\lambda\epsilon} + \frac{\log^2 n \cdot \log(1/\delta)}{\beta^2 \cdot \min(1, \epsilon^2)} \right).$$

In addition, all discussed bound hold across all events with probability at least $1 - n^{-2}$. This concludes the analysis. $\qquad \square$

## B.3 Proof of Lemma 5.4

**Lemma 5.4.** *Consider all lights vertices defined in Line 4 of Algorithm 1. Assume that $5\beta + 2\lambda < 1/1.1$. Then, with probability at least $1 - n^{-2}$, making as singleton clusters any subset of those light vertices increases the cost of clustering by $O(\text{OPT}/(\beta \cdot \lambda)^2)$, where $\text{OPT}$ denotes the cost of the optimum clustering for the input graph.*

*Proof.* Consider first a single light vertex $v$ which is not a singleton cluster. Let $C$ be the cluster of $\hat{G}'$ that $v$ initially belongs to. We consider two cases. First, recall that from our proof of Lemma 5.3 that, with probability at least $1 - n^{-2}$, we have that $0.9\lambda \leq \overline{\lambda}_v \leq 1.1\lambda$ and $0.9\beta \leq \overline{\beta}_{u,v} \leq 1.1\beta$, where $\overline{\lambda}$ and $\overline{\beta}$ are inputs to ALG-CC.

**Case 1: $v$ has at least $\overline{\lambda}_v/2$ fraction of neighbors outside $C$.** In this case, the cost of having $v$ in $C$ is already at least $d(v) \cdot \overline{\lambda}_v/2 \geq d(v) \cdot 0.9 \cdot \lambda/2$, while having $v$ as a singleton has cost $d(v)$.

**Case 2: $v$ has less than $\overline{\lambda}_v/2$ fraction of neighbors outside $C$.** Since $v$ is not in agreement with at least $\overline{\lambda}_v$ fraction of its neighbors, this case implies that at least $\overline{\lambda}_v/2 \geq 0.9 \cdot \lambda/2$ fraction of those neighbors are in $C$. We now develop a charging arguments to derive the advertised approximation.

Let $x \in C$ be a vertex that $v$ is not in a agreement with. Then, for a fixed $x$ and $v$ in *the same* cluster of $\hat{G}'$, there are at least $O(d(v)\beta)$ vertices $z$ (incident to $x$ or $v$, but not to the other vertex) that the current clustering is paying for. In other words, the current clustering is paying for edges of the form $\{z, x\}$ and $\{z, v\}$; as a remark, $z$ does not have to belong to $C$. Let $Z(v)$ denote the *multiset* of all such edges for a given vertex $v$. We charge each edge in $Z(v)$ by $O(1/(\beta\lambda))$.

On the other hand, making $v$ a singleton increases the cost of clustering by at most $d(v)$. We now want to argue that there is enough charging so that we can distribute the cost $d(v)$ (for making $v$ a singleton cluster) over $Z(v)$ and, moreover, do that for all light vertices $v$ simultaneously. There are at least $O(\beta \cdot d(v) \cdot \lambda \cdot d(v))$ edges in $Z(v)$; recall that $Z(v)$ is a multiset. We distribute uniformly the cost $d(v)$ (for making $v$ a singleton) across $Z(v)$, incurring $O(1/(\beta \cdot \lambda \cdot d(v)))$ cost per an element of $Z(v)$.

Now it remains to comment on how many times an edge appears in the union of all $Z(\cdot)$ multisets. Edge $z_e = \{x, y\}$ in included in $Z(\cdot)$ when $x$ and its neighbor, or $y$ and its neighbor are considered. Moreover, those neighbors belong to the same cluster of $\hat{G}'$ and hence have similar degrees (i.e., as shown in the proof of Lemma B.2, their degrees differ by at most $(1 - 4\beta)^2$ factor). Hence, an edge $z_e \in Z(v)$ appears $O(d(v))$ times across all $Z(\cdot)$, which concludes our analysis. $\qquad \square$

## C Lower bound

In this section we show that any private algorithm for correlation clustering must incur at least $\Omega(n)$ additive error in the approximation guarantee, regardless of its multiplicative approximation ratio. The following is a restatement of Theorem 1.2.

**Theorem C.1.** *Let $\mathcal{A}$ be an $(\epsilon, \delta)$-DP algorithm for correlation clustering on unweighted complete graphs, where $\epsilon \leq 1$ and $\delta \leq 0.1$. Then the expected cost of $\mathcal{A}$ is at least $n/20$, even when restricted to instances whose optimal cost is $0$.*

*Proof.* Fix an even number $n = 2m$ of vertices and consider the fixed perfect matching $(1, 2), (3, 4),$ $\ldots, (2m - 1, 2m)$. For every vector $\tau \in \{0, 1\}^m$ we consider the instance $I_\tau$ obtained by having plus-edges $(2i - 1, 2i)$ for those $i = 1, ..., m$ where $\tau_i = 1$ (and minus-edges for $i$ with $\tau_i = 0$, as well as everywhere outside this perfect matching). Note that this instance is a complete unweighted graph and has optimal cost $0$.

For $\tau \in \{0, 1\}^m$ and $i \in \{1, ..., m\}$ define $p_\tau^{(i)}$ to be the marginal probability that vertices $2i - 1$ and $2i$ are in the same cluster when $\mathcal{A}$ is run on the instance $I_\tau$.

Finally, for $\sigma \in \{0, 1\}^{m-1}$, $i \in \{1, ..., m\}$ and $b \in \{0, 1\}$ let $\sigma[i \leftarrow b]$ be the vector $\sigma$ with the bit $b$ inserted at the $i$-th position to obtain an $m$-dimensional vector (note that $\sigma$ is $(m - 1)$-dimensional). Note that $I_{\sigma[i \leftarrow 0]}$ and $I_{\sigma[i \leftarrow 1]}$ are adjacent instances. Thus $(\epsilon, \delta)$-privacy gives

$$p_{\sigma[i \leftarrow 1]}^{(i)} \leq e^\epsilon \cdot p_{\sigma[i \leftarrow 0]}^{(i)} + \delta \tag{18}$$

for all $i$ and $\sigma$.

Towards a contradiction assume that $\mathcal{A}$ achieves expected cost at most $0.05n = 0.1m$ on every instance $I_\tau$. In particular, the expected cost on the matching minus-edges is at most $0.1m$, i.e.,

$$0.1m \geq \sum_{i:\tau_i=0} p_\tau^{(i)}.$$

Summing this up over all vectors $\tau \in \{0, 1\}^m$ we get

$$2^m \cdot 0.1m \geq \sum_{\tau \in \{0,1\}^m} \sum_{i:\tau_i=0} p_\tau^{(i)} = \sum_i \sum_{\sigma \in \{0,1\}^{m-1}} p_{\sigma[i \leftarrow 0]}^{(i)} \tag{19}$$

and similarly since the expected cost on the matching plus-edges is at most $0.1m$, we get

$$
\begin{aligned}
2^m \cdot 0.1m &\geq \sum_{\tau \in \{0,1\}^m} \sum_{i:\tau_i=1} (1 - p_\tau^{(i)}) \\
&= \sum_i \sum_{\sigma \in \{0,1\}^{m-1}} (1 - p_{\sigma[i \leftarrow 1]}^{(i)}) \\
&\overset{(18)}{\geq} \sum_i \sum_{\sigma \in \{0,1\}^{m-1}} (1 - e^\epsilon \cdot p_{\sigma[i \leftarrow 0]}^{(i)} - \delta) \\
&= (1 - \delta) \cdot m \cdot 2^{m-1} - e^\epsilon \cdot \sum_i \sum_{\sigma \in \{0,1\}^{m-1}} p_{\sigma[i \leftarrow 0]}^{(i)} \\
&\overset{(19)}{\geq} (1 - \delta) \cdot m \cdot 2^{m-1} - e^\epsilon \cdot 2^m \cdot 0.1m \\
&\geq 0.45 \cdot m \cdot 2^m - 0.1e \cdot 2^m \cdot m.
\end{aligned}
$$

Dividing by $2^m \cdot m$ gives $0.1 \geq 0.45 - 0.1e$, which is a contradiction. $\qquad\square$