# OpenReview forum: "Near-Optimal Correlation Clustering with Privacy"
_NeurIPS.cc/2022/Conference — NeurIPS 2022 Accept_

### Official Review · Reviewer_d5Yt · 2022-07-07

**Rating:** 6
**Confidence:** 3
**Soundness:** 3 good
**Presentation:** 3 good
**Contribution:** 3 good

**Summary:**

The authors applied differential privacy to correlation clustering, using graph notations. They first added noise to the degree of vertices and discarded edges using the definition of noised agreement. The authors added noise to the number of edges and defined a vertex is light or heavy. Finally, they discarded the edge and output the heavy vertices in a cluster. Also, the authors proposed approximated method.

**Questions:**

I’ve never seen the term “(0,0)-differential private” before(In the proof of Lemma 4.2). Is it widely used? If not, what about using post-processing or saying "it does not spend privacy budget"?

**Limitations:**

The authors assumed every weight of an edge is -1 or  1.

**Strengths And Weaknesses:**

[Strengths]
- The authors carefully analyzed to support their argument, and provided time complexity.
- The authors applied differential privacy to recent work, especially using noised agreement have novelty.

[Weaknesses]
- There are some minor issues in the paper. For example, in lines 20-21, the authors need to say the sum of the “absolute value” of negative edges to prevent confusion.
- It seems that the privacy budget is wrongly calculated. In Line 1 of Algorithm 1, the authors added noise to every d(v). Since d(v) = |N(v)| where N is the set of neighbors of a vertex v, d(v)s(for v in V) are not independent, the privacy budget is accumulated by composition theorem. This implies the privacy budget is higher than the authors argued, exactly |G|/2 times higher than expected(The sensitivity of d is 2, as mentioned in line 227, and the sensitivity of d(v) is 1).
- It would be better if the authors change the arXiv references in the final submission. For example, [CALM+21], which appears to have motivated the authors is accepted by ICML.

---

> ### Author Response · Authors · 2022-08-02
> **Addressing the concerns and comments**
>
> Thank you for your review. We will apply the suggestions provided in the first and the third bullet under [Weaknesses].
>
>
> Regarding the comment on the wrong calculation of the privacy budget:
> Our calculation of the privacy budget in Lemma 4.1 is correct. We directly invoke Theorem 2.6, which holds for multivariate functions; we apply it to the multivariate function $d : \mathbb{G} \rightarrow \mathbb{R}^k$, where $k = |V|$ is the number of vertices of the graph. The crucial parameter in this theorem is the sensitivity of the function, which is defined as the maximum $\ell_1$-distance between $d(G)$ and $d(G’)$ for two adjacent datasets $G$, $G’$. In this paper, $G$ and $G’$ are adjacent if they differ by one edge. Adding an edge to a graph causes the degree of two vertices (its endpoints) to increment by one and does not change the degrees of other vertices. Thus the sensitivity of the function $d$ is 2.
>
> We do not employ any composition theorem to argue about the privacy budget for the noised degree sequence. However, if one wanted to do that, then, having fixed the datasets $G$ and $G’$ that differ in one edge $(x,y)$, one could say that the sensitivities of (univariate functions) $d(x)$ and $d(y)$ are 1 whereas the sensitivity of $d(v)$ for $v$ other than $x, y$ is 0. For any adjacent graphs $G$ and $G’$, there are only 2 vertices that are consuming the privacy budget, not $|V|$ many.
>
>
> Regarding the Question:
> By “(0,0)-differentially private” we essentially mean that the distributions of the outputs on two adjacent inputs are identical. Your proposed alternative wordings could also be used.
>
> Regarding Limitations:
> We are indeed considering unweighted complete graphs in this paper. The value -1 of an edge {u, v} means that u and v are dissimilar, while +1 means that u and v are similar. Please also see our response to Reviewer rmrh.

---

> > ### Comment · Reviewer_d5Yt · 2022-08-03
> > **Raised the score.**
> >
> > Thank you for your response.
> > I thought G and G' differ by one "node", but as you wrote in the comments and paper, they differ by one "edge", so the sensitivity of the lemma is correctly calculated.
> > I raise the score delightfully.

---

### Official Review · Reviewer_SoYq · 2022-07-11

**Rating:** 7
**Confidence:** 3
**Soundness:** 4 excellent
**Presentation:** 4 excellent
**Contribution:** 4 excellent

**Summary:**

The paper discusses the min-disagree correlation clustering problem. It gives an (\eps, \delta)-differentially private algorithm that provides O(1) multiplicative and O(n log^2n) additive approximation. The additive approximation is close to the optimal bound of \Omega(n), which is also shown in the paper. Theorems 1.1 and 1.2 appropriately summarise the main results of the paper.

**Questions:**

Even though the theoretical results in the paper are appreciated, it may be nice to have a discussion on the applied aspects of the results. Are there specific application scenarios where the results of this paper may be useful?

**Limitations:**

The work is theoretical. I do not see any negative societal impact.

**Strengths And Weaknesses:**

Strengths:
- The paper gives an improvement over previous work, improving the additive approximation guarantee.
- The lower bound result also improves the previously known results.
- The intuition behind the algorithm and the analysis is presented well.


Weaknesses:
- I do not find any significant weakness in the paper. One aspect that may not appeal to practitioners is that the results in this paper are theoretical. There are no experimental evaluations presented.

---

> ### Author Response · Authors · 2022-08-02
> **Acknowledge the comment**
>
> Thank you for your review. Please see our response to Reviewer Kj4P.

---

### Official Review · Reviewer_rmrh · 2022-07-11

**Rating:** 7
**Confidence:** 3
**Soundness:** 3 good
**Presentation:** 3 good
**Contribution:** 3 good

**Summary:**

The authors develop an approximation algorithm for correlation clustering in complete unweighted graphs with privacy guarantees. The additive error of their algorithm is stronger than prior work, while it is optimal up to polylogarithmic factors. It also matches the additive error of the exponential mechanism up to a logn factor. They also show that linear error is necessary even for complete unweighted graphs and (eps,delta)-privacy.

**Questions:**

-

**Limitations:**

No negative societal impact, as far as I could check.

**Strengths And Weaknesses:**

Strenghts:
- the paper is very well written and organized
- the additive error of their algorithm is stronger than prior work and optimal, up to polylog factors. To this end, the authors also prove that linear error is necessary even for complete unweighted graphs.
- the authors borrow some ideas from [CALM21], however, a deeper analysis of the algorithm in [CALM21] and several novel ideas are developed.

Weaknesses:
- their results seem to only apply to complete unweighted graphs (in contrast to [BEK21] and [LIU22] )
- some of the ideas were presented in [CALM21], however, the paper contains significant novel contributions

---

> ### Author Response · Authors · 2022-08-02
> **Agree with both points.**
>
> Thank you for your review. We agree with both points. We agree that finding a DP algorithm with optimal additive guarantee for weighted inputs is an interesting follow-up question. We note that, as exemplified in [CALM+, ICML21], the unweighted case already has numerous applications.

---

### Official Review · Reviewer_Kj4P · 2022-07-11

**Rating:** 7
**Confidence:** 3
**Soundness:** 4 excellent
**Presentation:** 4 excellent
**Contribution:** 3 good

**Summary:**

This paper studies the correlation clustering problem under edge differential privacy constraints. In the correlation clustering problem, given a complete graph with + and – labels on the edges, the goal is to partition the vertices so as to minimize the number of + edges between parts, plus the – edges inside parts. With edge differential privacy (DP), Given two graphs on the same set of vertices that differ in only one edge, the goal is to make sure, for any set of outcomes of an algorithm, the probability of returning that outcome is roughly the same (within a multiplicative $e^\epsilon$ and additive $\delta$) for both graphs. This problem was studied by BEK21 where they give an $(\epsilon, \delta)$-DP algorithm that approximates the optimal clustering cost (OPT) with cost $O(1) OPT + O(n^{1.75}log(n/\delta)/\epsilon)$. There is also a concurrent paper of Liu22 with the additive term of $\tilde{O}(n^{1.5}log(n/\delta)/\sqrt{\epsilon})$. BEK21 also prove an $\Omega(n)$ lowerbound on the additive approximation of DP algorithms on path graphs.

This paper improves the additive term to $O(nlog^2(n)log(n/\delta)/{\epsilon}^2)$. Inspired by CALM+21’s paper on correlation clustering in the parallel setting, they make the steps in that algorithm private in such a way to make sure the cost approximation remains within the mentioned bound. They also add to the lowerbound results by proving a linear lower bound on the approximation in complete unweighted graphs.


**Questions:**

As mentioned above, I recommend the authors elaborate on various aspects of the performance of their algorithm.

**Limitations:**

I do not see any serious limitations.

**Strengths And Weaknesses:**

The problem considered in this paper is natural and important. The paper is very well written, and the ideas are clearly presented. Although I did not check all the proofs in detail, the logic seems solid to me. The paper borrows ideas from the literature, but they had to do nontrivial work to prove the DP and cost guarantees.

My only comment for the authors is to elaborate on how well the algorithm scales. How parallelizable it is, or maybe bring empirical studies to argue about this. In general, this paper could benefit from empirical studies on how well the algorithm actually performs in practice. For example, the dependence on $\epsilon$ is quadratic, what are some ranges of $\epsilon$ that are used in practice. Or in general, please elaborate on the significance of your result in the real-world setting.

---

> ### Author Response · Authors · 2022-08-02
> **Acknowledging the comment**
>
> Thank you for your review. We agree that it is an interesting open question how to obtain an efficient and scalable DP correlation clustering algorithm. Concerning the running time, we note that our work does not require solving an exponential-size LP as required by the approach of [BEK, ICML21] and is inspired by the approach of [CALM+, ICML21], for which the authors have shown solid practical results (including that it performs very well in a massively parallel setting). A thorough experimental study of our algorithm and designing a parallel nearly-optimal differentially private algorithm is an interesting follow-up direction.

---

> > ### Comment · Reviewer_Kj4P · 2022-08-08
> > **Response**
> >
> > I thank the authors for the response and would like to see this added to the paper, maybe a few lines in the introduction.

---

> > > ### Author Response · Authors · 2022-08-08
> > > **Response**
> > >
> > > Thanks for your comment, we will extend the introduction and add these.

---

### Meta-Review · Area_Chair_KZuR · 2022-08-25

**Recommendation:** Accept
**Confidence:** Certain

**Metareview:**

The paper gives a new algorithm for correlation clustering in the edge differential privacy model i.e. two graphs are neighbor if they differ by one edge. The problem has been studied extensively in the nonprivate setting and there are several previous works on the problem in the private setting. The new algorithm has nearly optimal additive error and it is quite simple and efficient. The new additive error is nearly linear in n, the number of nodes, as opposed to n^1.5 in the previous polynomial time algorithms. All reviewers appreciate the near optimal solution to a well studied problem. The reviewers highly value the theoretical contribution but also suggest that the paper can be strengthened with some empirical evaluation.

**Award:**

No

---

### Decision · Program_Chairs · 2022-09-14

Accept